# Rab35 governs apicobasal polarity through regulation of actin dynamics during sprouting angiogenesis

Caitlin R. Francis[1], Hayle Kincross[1] & Erich J. Kushner ●[1] ✉

In early blood vessel development, trafficking programs, such as those using Rab GTPases, are tasked with delivering vesicular cargo with high spatiotemporal accuracy. However, the function of many Rab trafficking proteins remain ill-defined in endothelial tissue; therefore, their relevance to blood vessel development is unknown. Rab35 has been shown to play an enigmatic role in cellular behaviors which differs greatly between tissue-type and organism. Importantly, Rab35 has never been characterized for its potential contribution in sprouting angiogenesis; thus, our goal was to map Rab35's primary function in angiogenesis. Our results demonstrate that Rab35 is critical for sprout formation; in its absence, apicobasal polarity is entirely lost in vitro and in vivo. To determine mechanism, we systematically explored established Rab35 effectors and show that none are operative in endothelial cells. However, we find that Rab35 partners with DENNd1c, an evolutionarily divergent guanine exchange factor, to localize to actin. Here, Rab35 regulates actin polymerization through limiting Rac1 and RhoA activity, which is required to set up proper apicobasal polarity during sprout formation. Our findings establish that Rab35 is a potent brake of actin remodeling during blood vessel development.

Angiogenesis is the process of sprouting and growth of new blood vessels from preexisting ones and is the primary driver of network expansion[1–4]. Many extrinsic and intrinsic biological systems have been shown to affect endothelial biology and, by extension, blood vessel formation. Membrane trafficking is one such system that is less well-characterized in endothelial tissue but has recently become more appreciated as additional organotypic trafficking signatures are aligned with important endothelial behaviors[5–8]. Membrane trafficking refers to vesicular transport of protein(s) to, or in vicinity of, the plasma membrane[9–11]. Here, trafficking regulators, such as Rab GTPases, interface with a host of effectors involved in receptor recycling, cytoskeletal regulation, shunting to degradative organelles, lumen formation, basement membrane secretion, and many other signaling events[9,12,13]. Indeed, critical to the understanding of how endothelial cells build dynamic and resilient vascular structures is the regulation of membrane trafficking during angiogenic development[14].

The GTPase Rab35 has been shown to be a multi-faceted regulator of membrane trafficking and continues to be an intensely researched Rab family member[15]. The promiscuity of Rab35 touching multiple pathways has created a cognitive bottleneck in attempting to assign function in any system, due to its seemingly endless diversity of roles. For instance, Rab35 has been shown to be involved in cytokinesis as well as transcytosis of the apical protein podocalyxin during lumen biogenesis in epithelial cysts[16,17]. In other investigations, Rab35 has been reported to be a negative regulator of the integrin recycling protein Arf6 via its effector ACAP2[18–20]. Additionally, MICAL-1 has been shown to also facilitate Rab35's association with Arf6 and play a role in actin turnover[20–22]. In Drosophila, Rab35 regulates apical constriction during germband extension as well as actin bundling via recruitment of Fascin[23,24]. To date, there is no unified study on Rab35 taking into account its many disparate functions in any tissue. Regarding blood vessel function,

[1]Department of Biological Sciences, University of Denver, Denver, CO, USA. ✉e-mail: Erich.Kushner@du.edu

no endothelial studies exist detailing how, or if, Rab35 functions in sprouting angiogenesis.

In the current study, our goal was to comprehensively characterize Rab35's role in sprouting angiogenesis. To do so, we took a holistic approach in investigating established partners of Rab35 and characterized their effect on sprouting behaviors and downstream cellular morphodynamics in vitro and in vivo. Primarily using a three-dimensional sprouting assay, our results reveal that Rab35 is required for sprouting as its loss significantly disrupts apicobasal polarity. Focusing on Rab35 effectors, we demonstrate that of the many reported effectors only ACAP2 is capable of directly binding Rab35 in endothelial cells. However, upon investigating ACAP2 and its target Arf6, we determine this established Rab35 trafficking cascade is largely insignificant with regard to sprouting angiogenesis. Excluding all other pathways, we focused on the Rab35 guanine exchange factor (GEF), DENNd1c, and its role in localizing Rab35 to actin structures. Our results demonstrate that DENNd1c facilitates Rab35 tethering to the actin cytoskeleton. Once on actin, Rab35 acts as a negative regulator of actin polymerization and is critical for the formation of proper actin architecture. In vivo, we show the requirement of Rab35 in zebrafish blood vessel development using a gene editing approach. Overall, our results provide evidence of a focused role for Rab35 as a regulator of actin assembly during sprouting angiogenesis.

## Results

### Rab35 is required for sprouting angiogenesis

To characterize the role of Rab35 in sprouting angiogenesis, we first cloned a fluorescently tagged version of Rab35 into a lentivirus expression system[25]. Thereafter, we transduced ECs and then challenged the cells to sprout in a fibrin-bead assay (Fig. 1a)[26,27]. Rab35 in 3-dimensional (3D) sprouts demonstrated strong membrane localization, co localizing with apical marker podocalyxin and luminal actin, opposite basally located β1-integrin (Fig. 1b and Supplementary Movie 1). To test whether Rab35 was necessary for endothelial sprouting, we knocked down Rab35 using siRNA (Fig. 1c). Loss of Rab35 reduced sprout length and sprouts per bead by ~50%, with a significant increase in the percentage of non-lumenized sprouts (Fig. 1d–g). Morphologically, the sprouts appeared stubby, non-lumenized, and generally dysmorphic compared with controls (Fig. 1d). These results indicate that Rab35 is required for proper sprout development.

Given Rab35 depletion exhibited such a profound impact on sprouting parameters, we stained for various cytoskeletal, apical, and basal markers to determine if Rab35 was affecting specific polarity pathways or producing a more global cellular defect. Imaging for VE-cadherin (cell-cell junctions), podocalyxin, β1-integrin (basal membrane), moesin (cytoskeletal, apical membrane), synaptotagmin-like protein-2 (apical membrane), and phosphorylated-Tie2 (apical membrane) revealed that Rab35 knockdown affected all protein localization (Supplementary Fig. 1A), suggesting that loss of Rab35 globally disturbs cell polarity programs. Emblematic of this was the significant lack of lumen formation and the increase in discontinuous vacuoles in the Rab35 depleted condition (Supplementary Fig. 1B), as lumenogenesis requires proper apicobasal signaling to form[7,28]. We also observed that Rab35 knockdown reduced the number of nuclei per sprout, indicating the presence of cell division defects in line with other reports[17,29–31] (Supplementary Fig. 1C). We tested for Rab35 knockdown efficiency in prolonged 2D culture to ensure our siRNA knockdown was not diminished at day 4 of sprouting. At day-4 and day-5, Rab35 expression was dramatically lower than controls (Supplementary Fig. 1D). Overall, this data suggests that Rab35 plays a significant role in establishing cell polarity during angiogenic sprouting.

Using a mosaic approach, we determined the cell autonomous nature of Rab35 depletion in a sprout collective. To do so, we treated ECs with either Rab35 siRNA or a scramble control and then mixed 50:50 with wild-type ECs. The resulting mosaic sprouts contained a mixture of siRNA-treated and untreated ECs (Fig. 1h). Cells contained within sprouts were then binned into two categories: (1) not-opposing, an isolated siRNA-treated cell; or (2) opposing, two adjacent siRNA-treated ECs (Fig. 1i, j). Rab35 knockdown in not-opposing ECs contained actin-labeled vacuolations and polarity defects as indicated by a reduction in lumen formation compared with scramble-treated controls (Fig. 1k–m). For Rab35 depleted ECs in the opposing orientation, defects were more pronounced with complete lumen failures at these sites, while also exhibiting multiple vacuolations and polarity defects (Fig. 1m). Overall, these results indicate that Rab35 is cell autonomous and is required for EC polarity.

### Rab35 resides at the apical membrane during sprouting

Next, we sought to better understand Rab35's cellular localization to gain insight into its potential function. In sprouts, quantification of Rab35 enrichment between different cellular compartments showed a preference for the apical membrane for wild-type (WT) and constitutively active (CA) Rab35 variants, while the dominant-negative (DN) Rab35 mutant resided in the cytoplasm (Fig. 2a). Subcellular imaging of WT and CA Rab35 showed a strong colocalization with apical podocalyxin (Fig. 2b). Similar to loss of Rab35, expression of the DN Rab35 also produced polarity defects, such as mislocalization of podocalyxin and large actin accumulations (Fig. 2b). To more conclusively assign Rab35 phenotypes, we performed several rescue assays by knocking down the endogenous Rab35 protein and then over-expressing Rab35 variants in sprouts. Expression of WT or CA Rab35 decreased the number of non-lumenized sites in sprouts compared to Rab35 knockdown alone expressing a GFP control, but not to levels in the scramble treated group (Fig. 2c, d; Supplementary Fig. 2A, B). Rab35 knockdown and expression of the DN Rab35 mutant showed the highest increase in dysmorphic sprouts, exhibiting numerous accumulations of actin puncta and lumen defects, again suggesting Rab35 is associated with sprout function.

Within the sprout body, Rab35 also localized to actin at cytokinetic bridges as previously described[17,29,31], but had no preference for filopodia extensions or tip-cell positioning (Supplementary Fig. 3A, B; Supplementary Fig. 4A–C). We also observed that Rab35 modestly colocalized with filamentous actin in a monolayer; however, this association was reduced in migratory cells (Supplementary Fig. 3C, D). Previous reports have implicated Rab35 in Wiebel Palade Body (WPB) granule release[32]; although, in our hands, Rab35 did not colocalize with these structures in 2D or 3D culture systems (Supplementary Fig. 3E, F). These results indicate that Rab35 is largely localized to the apical membrane in its active form as well as areas of high actin density.

Reports in epithelial tissue suggest that Rab35 participates in trafficking of podocalyxin to the apical membrane[16,17]. In the sprouting model, we indeed observed a strong colocalization of Rab35 and podocalyxin at the apical membrane as well as mislocalization of podocalyxin in the absence of Rab35. This data could be interpreted as a loss of, or defective, podocalyxin trafficking given Rab35's previous association with this pathway. As colocalization of podocalyxin and Rab35 at the apical membrane could be circumstantial as numerous proteins localize to the apical membrane during lumenogenesis, we overexpressed TagRFP-Rab35 and stained for endogenous podocalyxin in 2D culture and did not detect any significant signal overlap (Supplementary Fig. 5A). Previous literature showed that Rab35 directly binds to the cytoplasmic tail of podocalyxin[17]. Overexpression of the human podocalyxin cytoplasmic domain (residues 476–551) and TagRFP-Rab35 also did not show any obvious association (Supplementary Fig. 5A). To further probe for binding between Rab35 and podocalyxin, we engineered a mitochondrial-targeted (Tom20) Rab35 to test what proteins or complexes bind Rab35 and are then 'pulled' along to mitochondria. Expression of WT or CA Tom20-Rab35 did not show any association with endogenous podocalyxin or overexpression of its cytoplasmic tail domain (Supplementary Fig. 5B, C). We next

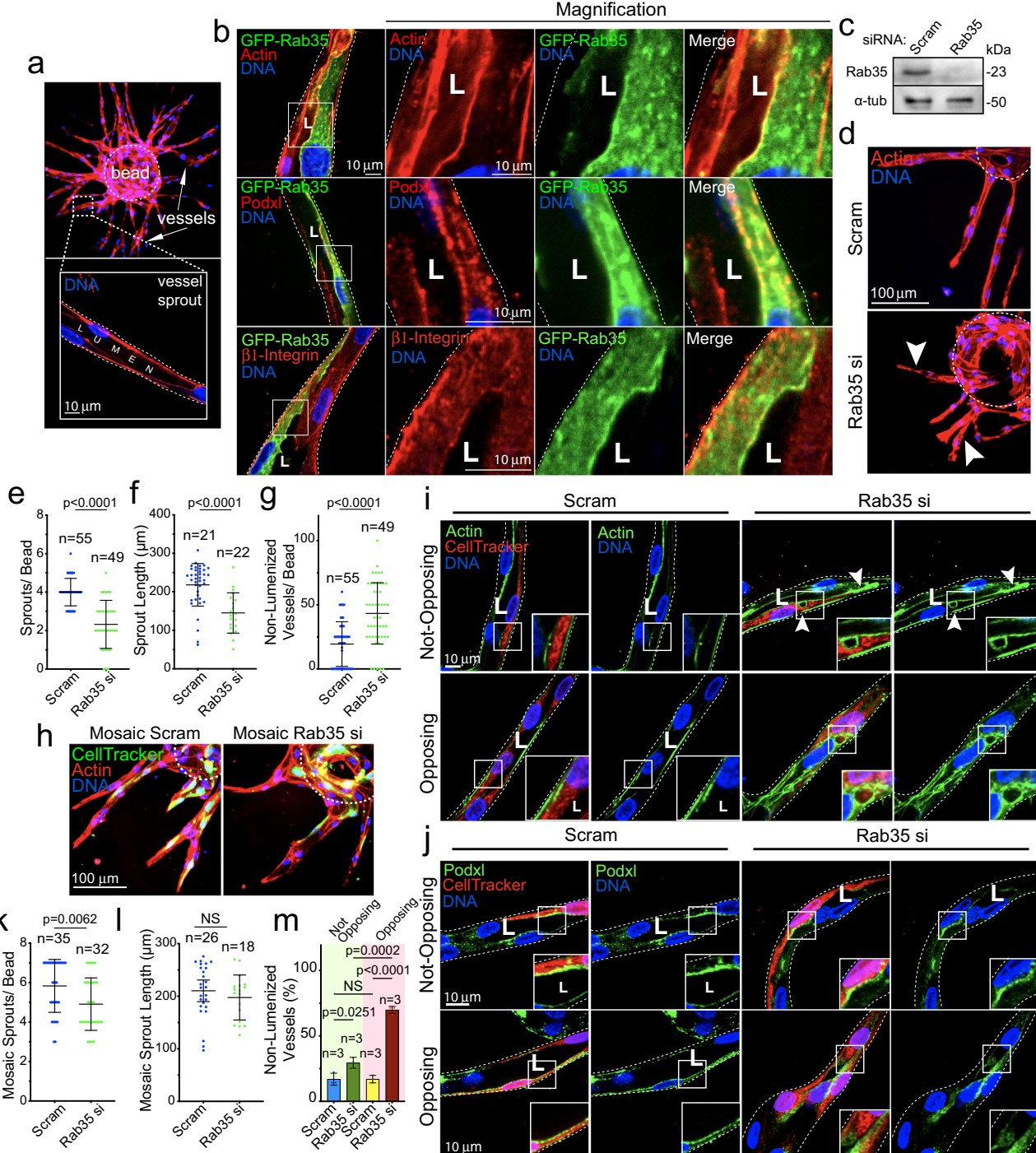

**Fig. 1 | Rab35 is an apical membrane protein required for sprout formation.**
**a** Representative images of the fibrin-bead assay (FBA) at low and high magnification. Arrows mark sprout structures. Inset depicts lumenized sprout. **b** GFP-Rab35 localization in endothelial sprouts with actin (top panels), podocalyxin (Podxl, middle panels), and β1-integrin (bottom panels). **c** Western blot confirmation of siRNA (si) knockdown (KD) of Rab35 (average 72.5% KD relative to control, *n* = 3). **d** Representative image of scramble (Scram) control and Rab35 siRNA KD sprouts. Arrowheads denote short and non-lumenized sprouts. Dashed lines outline the microbead. **e–g** Graphs of indicated sprouting parameters between groups. *n* = number of sprouts. Error bars represent standard deviation, middle bars are the mean. **h** Representative images of sprout morphology of mosaic Scram and Rab35 KD cells, green indicates cell tracker of siRNA treated cells. **i, j** Representative

images of non-opposing (top panels, an isolated siRNA treated cell) and opposing (bottom panels, two adjacent siRNA treated cells) cells stained as indicated. Arrowheads denote aberrant actin accumulations. **k, l** Quantification of indicated parameters across groups. *n* = number of sprouts. Error bars represent standard deviation, middle bars are the mean. **m** Quantification of non-lumenized sprout area across indicated groups. *n* = mean percentage of each experimental repeat (each group contains >20 cells). Error bars represent standard deviation, middle bars are the mean. In all images L denotes lumen. NS = non-significant. Statistical significance was assessed with an unpaired t-test or a 1-way ANOVA followed by a Dunnett multiple comparisons test. Insets are areas of higher magnification. White dotted lines mark sprout exterior. All experiments were done using Human umbilical vein endothelial cells in triplicate.

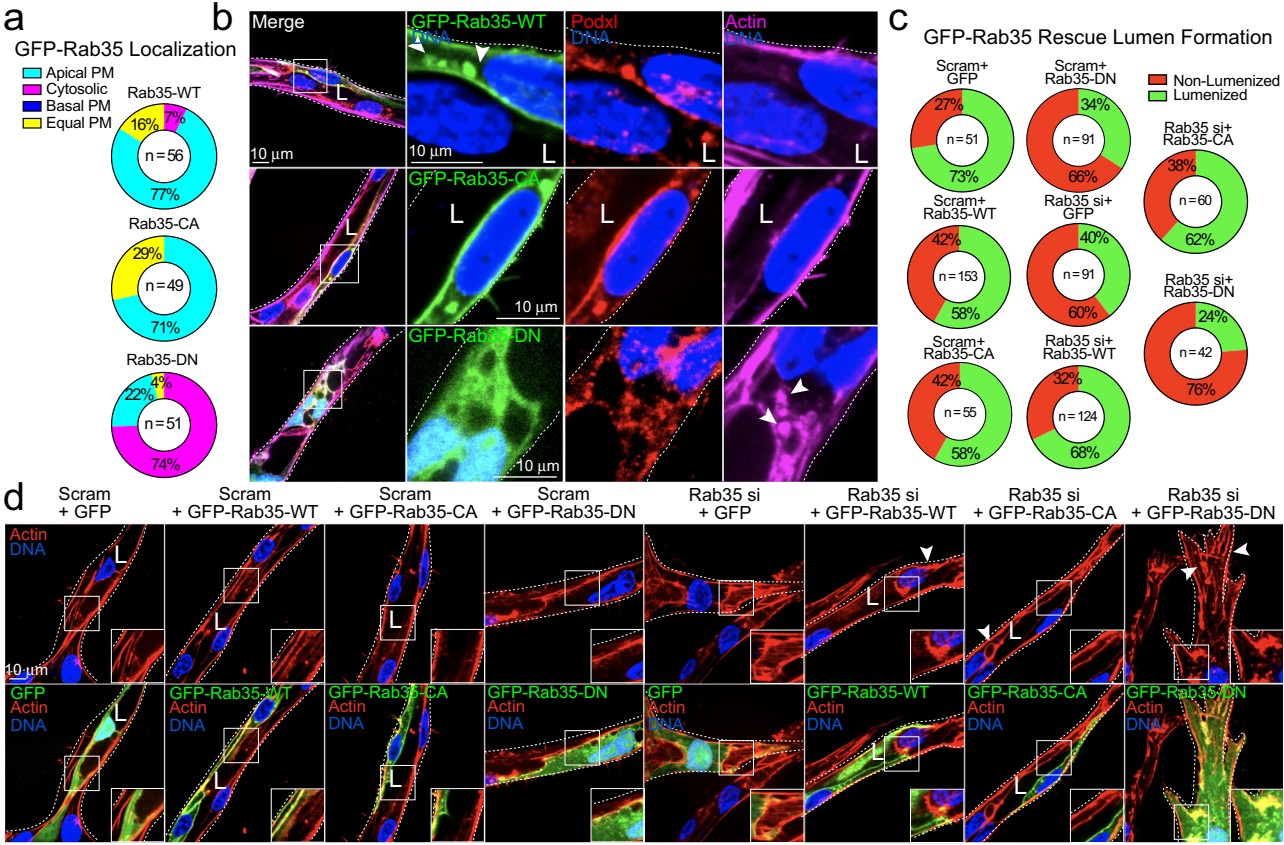

**Fig. 2 | Rab35 mutant localization and rescue in endothelial sprouts.**
**a** Quantification of GFP-Rab35 wild-type (WT), constitutively-active (CA), and dominant-negative (DN) localization in endothelial sprouts. Apical plasma membrane (PM, uniformly localized to apical membrane), basal PM (Rab35 uniformly located at the basal membrane), cytosolic (localized in the cytoplasm), equal PM (Rab35 equally distributed between the apical and basal membranes). *n* = number of cells. **b** GFP-Rab35 WT (top panels), CA (middle panels), and DN (bottom panels) localization in endothelial sprouts. Co-staining with podocalyxin (Podxl) and actin. Arrowheads in top panels denote Rab35 apical localization and puncta. Arrowheads

in bottom panels denote abnormal accumulations of actin. **c** Rescue experiment using scrambled (Scram) or Rab35 siRNA (si)-mediated knockdown (KD) with overexpression of indicated constructs. Percentages represent quantification of lumen formation in described conditions. *n* = number of sprouts. **d** Representative images of Scram and Rab35 KD sprouts expressing either GFP (control), or GFP-Rab35-WT/CA/DN. Arrowheads denote actin accumulations. White dotted lines mark sprout exterior. L denotes lumen in all images. Insets are areas of higher magnification. All experiments were done using human umbilical vein endothelial cells in triplicate.

reasoned if mistrafficking of podocalyxin by way of Rab35 depletion was the predominant mechanism underpinning the sprouting defects, then knocking down podocalyxin would produce a similar phenotype as compared with loss of Rab35. Knockdown of podocalyxin did not phenocopy Rab35-mediated sprouting defects (Supplementary Fig. 5D–I). The only exception was that podocalyxin knockdown increased the percentage of non-lumenized sprouts compared with controls. Overall, our data suggests that Rab35 does not directly participate in podocalyxin trafficking in ECs.

**Rab35 interacts with ACAP2 in endothelial cells**
To take a more holistic approach in determining how Rab35 functions in endothelial tissue, we performed a functional screen by knocking down the most highly cited Rab35 effectors (ACAP2, Rusc2, OCRL, MICAL-L1, MICAL-1, and Fascin) singly and in combination, to determine if any effector phenocopied Rab35 sprouting defects (Fig. 3a, b)[16,18–21,30,32–35]. First, we found that Rab35 itself did not produce a significant effect on 2D cell motility, suggesting the primary defect in sprouting may be due to altered apicobasal polarity only detectable in a 3D sprout environment (Supplementary Fig. 6A, B). As Rab35 and ACAP2 have been shown to affect the integrin recycling pathway via their association with Arf6, we also assayed for integrin recycling as defective integrin signaling could also affect cell polarity. As compared with the scramble controls, knockdown of Rab35, ACAP2, and MICAL-L1 significantly increased integrin recycling, while OCRL, MICAL-1, and

Fascin (treated with inhibitor NP-G2-044) had no effect or were directionally dissimilar (Supplementary Fig. 6C, D).

Next, we determined that Rusc2 protein levels were not detectable in ECs, thus was excluded from our screen (Supplementary Fig. 6E). Focusing on single knockdowns, ACAP2, OCRL, MICAL-L1, MICAL-1, and Fascin demonstrated phenotypic similarity to Rab35 knockdown with regard to sprouting parameters (Fig. 3c–f). Upon closer inspection, all proteins, excluding MICAL-L1, were associated with elevated frequencies of non-lumenized sprouts with varying degrees of disorganized actin; albeit, to a much a lesser extent than compared with Rab35 (Fig. 3g). Unlike loss of Rab35, individual knockdowns of all other proteins did not significantly reduce sprout length, suggesting a potential difference in phenotypes. For double knockdowns, we primarily focused on associations with ACAP2 as this is a more established Rab35 effector. Similar to individual knockdowns, loss of any combination of effectors significantly promoted lumen defects; although, no combination phenocopied Rab35 knockdown sprouting length defects (Supplementary Fig. 6F–I).

Both ACAP2 and OCRL have been reported to directly bind Rab35[18–20,30,32,35]; however, this interaction has not been validated in ECs. First, we overexpressed tagged versions of ACAP2, OCRL, MICAL-L1, MICAL-1, and Fascin to visualize their localization patterns with Rab35 in ECs. Only ACAP2 and Fascin showed strong colocalization with Rab35 at the plasma membrane along with peripheral actin (Fig. 4a, b). We again used the mitochondrial-targeted Tom20-Rab35

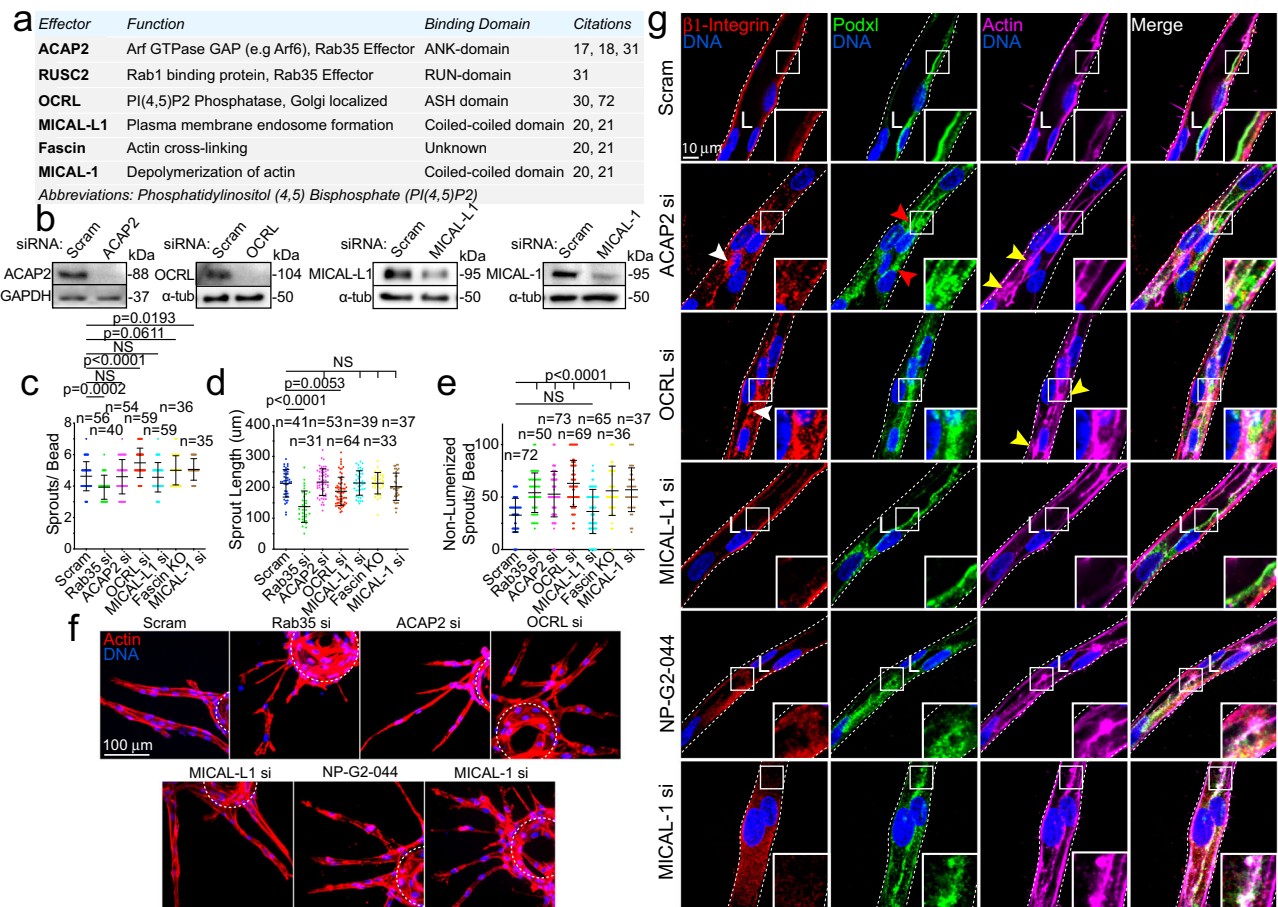

**Fig. 3 | Rab35 effector localization and requirement for sprouting. a** Table listing each effector, respective function, and citations. **b** ACAP2, OCRL, MICAL-L1 and MICAL-1 knockdown (KD) validation by western blotting (ACAP2 average 70.5% KD relative to control, $n = 3$; OCRL average 72.9% KD relative to control, $n = 3$; MICAL-L1 average 61.5% KD relative to control, $n = 3$; MICAL-1 average 69.3% KD relative to control, $n = 4$). **c**–**e** Graphs of indicated sprout parameters between groups. n=number of sprouts. Error bars represent standard deviation, middle bars are the mean. **f** Representative images of sprout morphology between indicated siRNA (si) KD groups. Dashed lines outline microbeads. **g** Representative images of siRNA-mediated KD of each effector. White arrowhead denotes abnormal localization of β1-integrin. Red arrowheads denote abnormal podocalyxin (Podxl) localization. Yellow arrowheads denote abnormal actin accumulations. White dotted lines mark sprout exterior. In all images L denotes lumen. NS = non-significant. Statistical significance was assessed with an unpaired t-test or a 1-way ANOVA followed by a Dunnett multiple comparisons test. $n$ = number of sprouts. Insets are areas of higher magnification. All experiments were done using human umbilical vein endothelial cells in triplicate.

to visualize any physical association between Rab35 and these previously published effectors. Co-expression of WT and CA Tom20-Rab35 with ACAP2 demonstrated strong colocalization at the mitochondria, while the DN Rab35 showed no significant binding of ACAP2 (Fig. 4c, d; Supplementary Fig. 7A, B). We performed this same experiment using ACAP2 with the ankyrin repeat domain deleted and observed no binding, indicating Rab35 directly interacts with this domain (Supplementary Fig. 7B). As a control, we also co-expressed a tom20-Rab27a and ACAP2 and observed no mislocalization of ACAP2 (Supplementary Fig. 7C). Co-expression of WT, CA or DN Tom20-Rab35 with OCRL, MICAL-L1, MICAL-1 or Fascin did not show any colocalization at the mitochondria, signifying a lack of binding (Fig. 4c, d; Supplementary Fig. 7D–H). These results demonstrate that only ACAP2 directly interacts with Rab35 in endothelial tissue.

## Rab35 activates Arf6 activity in endothelial cells

ACAP2 has been implicated as a GTPase activating protein (GAP) with Rab35 to inactivate the GTPase Arf6[18,19,33]. To test if this association exists in ECs, we first determined the localization of Arf6 relative to Rab35 and ACAP2 in culture. Cells expressing Rab35 and Arf6, or ACAP2 and Arf6 demonstrated modest colocalization primarily at the cell cortex (Supplementary Fig. 8A, B). Using WT, CA, and DN versions

of Arf6, we probed for actin to determine if Arf6, like Rab35, associated with these filaments. Wild-type and CA Arf6 demonstrated moderate colocalization with actin; however, this association did not persist on filamentous actin located towards the cell interior (Supplementary Fig. 8C). In sprouts, Arf6 showed modest localization to the apical membrane as compared to Rab35 (Supplementary Fig. 8D). Mitochondrial-targeted Rab35 or ACAP2 did not pull Arf6, indicating a lack of binding interaction (Supplementary Fig. 8E). We reasoned that the lack of binding between Arf6 and ACAP2 could be due to an insufficiency of Rab35; however, simultaneous expression of Tom20-GFP-Rab35 and TagRFP-ACAP2 did not localize HA-Arf6 to mitochondria, suggesting ACAP2 does not directly act upon Arf6, or that this signaling does not require a robust binding interaction in ECs (Supplementary Fig. 8F).

Due to the wealth of literature demonstrating loss of Rab35 increases Arf6 activity in non-endothelial tissues, we sought to confirm this signaling biochemically. First, we expressed WT, CA, and DN versions of Arf6 in ECs and used recombinant GGA3 to pulldown the active form of Arf6[36] (Supplementary Fig. 9A). Knockdown of Rab35 increased Arf6 activity as others have described[18,19,29] (Supplementary Fig. 9B, C). Given our mitochondrial-mistargeting results, this may be due to a more transient protein-protein interaction between the

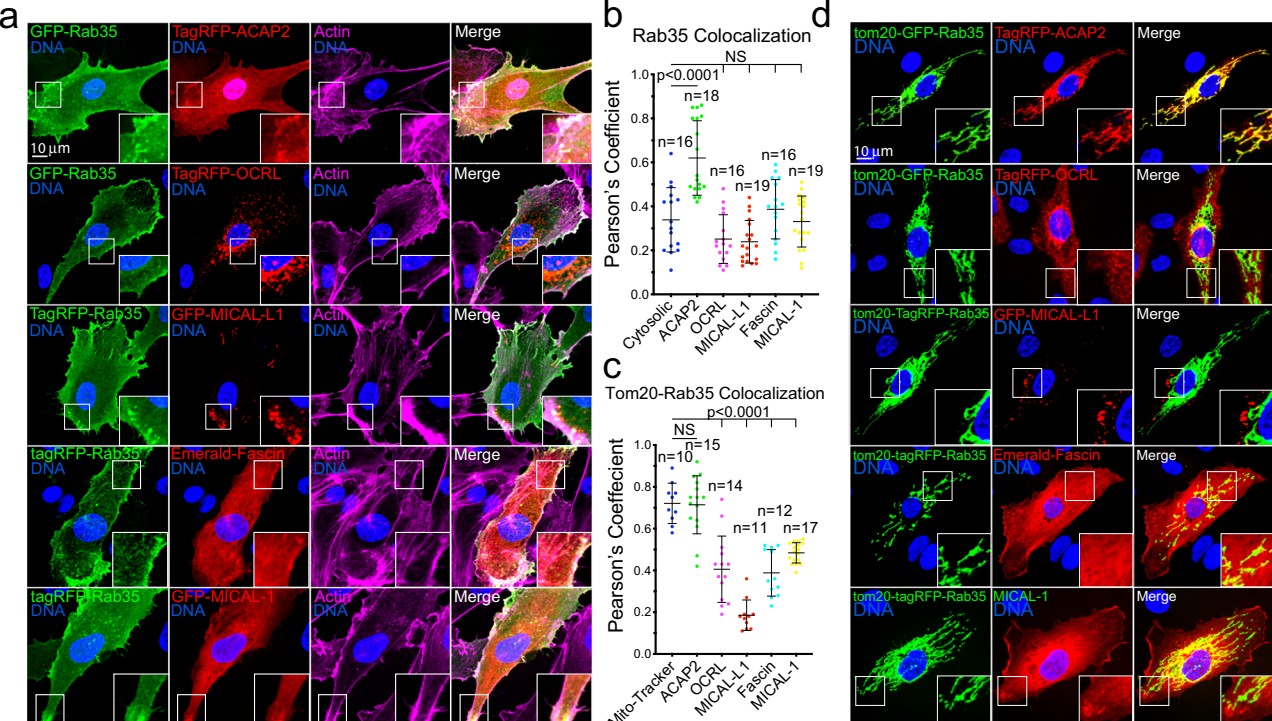

**Fig. 4 | ACAP2 binds with Rab35. a** Two-dimensional localization of GFP-Rab35 or TagRFP-Rab35 with indicated effectors and stained for actin. **b** Pearson's Coefficient between Rab35 and indicated effectors. *n* = number of cells. Error bars represent standard deviation, middle bars are the mean. **c** Pearson's Coefficient between Tom20-Rab35 and indicated effectors. n= number of cells. Error bars represent standard deviation, middle bars are the mean. **d** Representative images of mitochondrial mis-localization experiment. Rab35 was tethered to the mitochondria with a tom20 N-terminal tag to test if indicated effectors were also mislocalized. NS = non-significant. Statistical significance was assessed with an unpaired t-test or a 1-way ANOVA followed by a Dunnett multiple comparisons test. Insets are areas of higher magnification. All experiments were done using Human umbilical vein endothelial cells in triplicate.

Rab35/ACAP2 complex and Arf6, or potentially mediated through other unidentified Rab35 effectors. These results suggest that loss of Rab35 increases Arf6 activation; thus, we next tested if overactivation of Arf6 would phenocopy the Rab35 loss of function sprouting phenotype. We observed that overexpression of Arf6 marginally affected sprouting parameters with the WT and CA Arf6 increasing the frequency of lumen failures (Supplementary Fig. 9D). Inconsistent with Rab35 knockdown-associated actin aggregates and non-apical podocalyxin, ECs expressing CA Arf6 demonstrated normal actin architecture as well as typical polarity markers (Supplementary Fig. 8D). These results suggest that overactivation of Arf6 due to loss of Rab35 is likely not the causative pathway underlying sprouting defects.

Exploring the Arf6 axis further, we knocked down Arf6 to determine how this would affect sprouting parameters. Loss of Arf6 significantly increased the proportion of non-lumenized sprouts to a greater extent than Rab35 (Supplementary Fig. 9E–I). Interestingly, double knockdown of Rab35 and Arf6 did not further exacerbate this phenotype, suggesting that Arf6's effect on sprouting is independent from, or upstream of Rab35. Lastly, we tested if there was a dependency of Rab35 on Arf6, or vice versa, for proper localization in 3D sprouts. Knockdown of either protein did not prevent normal localization to apical actin (Supplementary Fig. 9J), suggesting Rab35's and Arf6's localization behaviors are likely not functionally linked during angiogenesis.

### DENNd1c is required for Rab35 function

We were intrigued by the idea that other roles of Rab35 were being unaccounted for as Arf6 overactivation could not fully reprise the Rab35 knockdown phenotype. To this end, Rab35 has three GEFs, DENNd1a-c[37–40]. DENNd1c has been shown to be not involved with GTP hydrolysis, but has the lone ability to bind to both globular and filamentous actin, potentially mediating Rab35 localization[38]. Exploring this association, we knocked down DENNd1a-c individually and in combination. Loss of DENNd1a and DENNd1b did not produce any significant impact on sprouting morphology; however, knockdown of DENNd1c alone resulted in growth of dysmorphic sprouts mirroring Rab35 loss of function (Fig. 5a–g; Supplementary Movies 2, 3). Knockdown of all DENNd1s produced the greatest effect on sprouting behaviors, presumably because the GEF activity provided by DENNd1a/b was also lost (Fig. 5d–f). Knocking down any given DENNd1 did not result in a compensatory increase in expression of the remaining DENNd1s (Supplementary Fig. 10A). We next cloned and tagged DENNd1c and confirmed colocalization with Rab35 on actin in 2D cell culture (Supplementary Fig. 10B, C). We also expressed Rab35 with the integral actin protein Arp2 known to mediate actin branching[41]. Rab35, Arp2, and filamentous (F)-actin strongly colocalized in the cell cortex (Supplementary Fig. 10B). To explore if DENNd1c, per se, was responsible for tethering Rab35 to actin, we individually knocked down all three DENNd1s and quantified the relative amount of Rab35 uniformly localized at the plasma membrane, accumulated at the plasma membrane or in the cytoplasm. DENNd1c knockdown exhibited the greatest increase in apical plasma membrane accumulations compared with DENNd1a or DENNd1b (Fig. 5h). These data indicate that loss of DENNd1c phenocopies the Rab35 knockdown effect on sprouting parameters and the actin cytoskeleton.

### Rab35 and DENNd1c localize to sites of actin polymerization

We next characterized the association between Rab35, DENNd1c, and branched actin. Both Rab35 and DENNd1c demonstrated strong colocalization to Arp2 and the underlying actin (Fig. 6a). To perturb the branched actin network, we next treated cells with the Arp2/3 inhibitor CK-666[42] and then determined the effect on Rab35 and DENNd1c

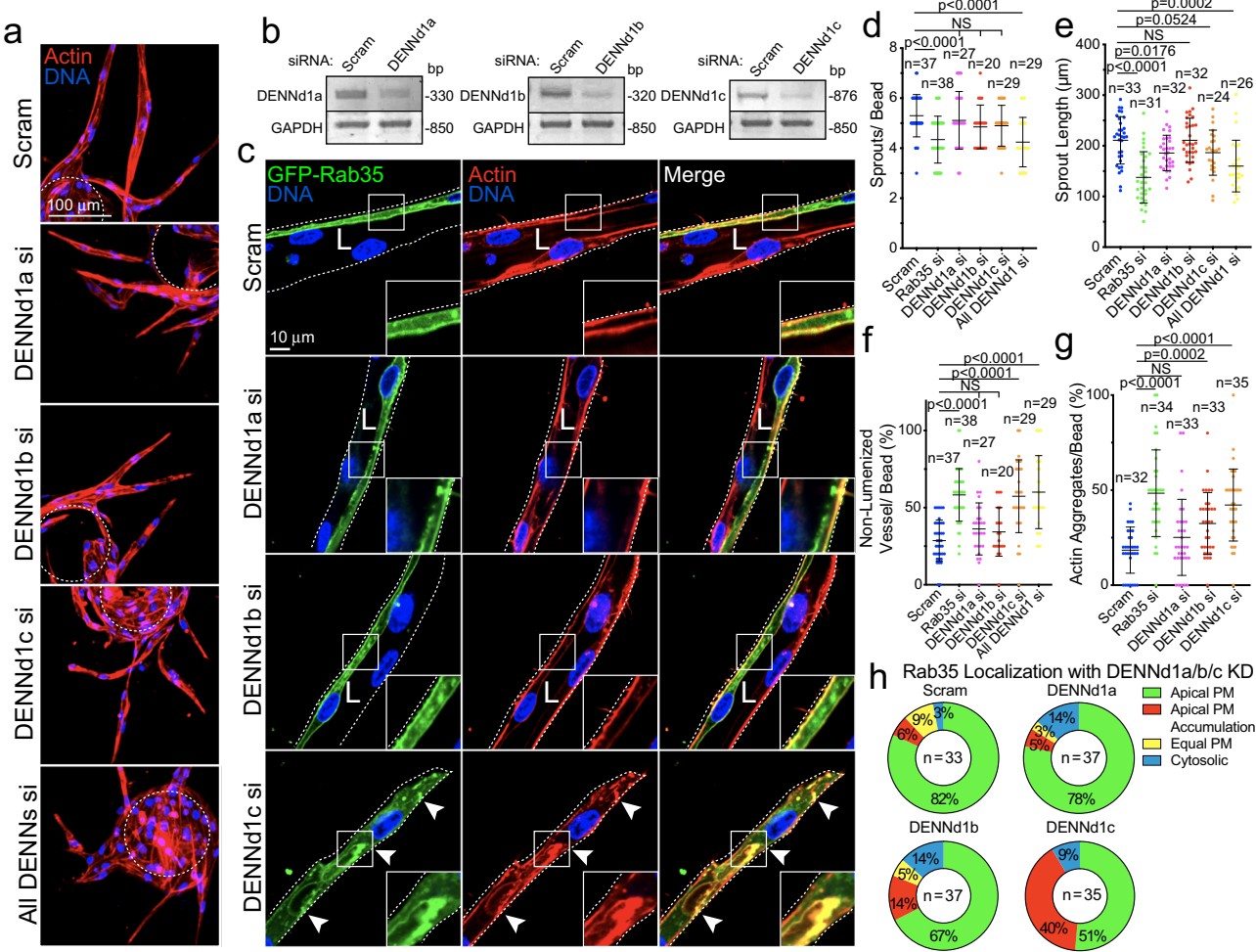

**Fig. 5 | DENNd1c is required for sprouting and Rab35 function. a** Sprout morphology of scramble (Scram), DENNd1a-c, and combined siRNA (si)-treated sprouts, stained with actin to denote the general morphology. Dashed line denotes microbead. **b** Knockdown confirmations for DENNd1a-c by RT-PCR. Base-pair (BP). **c** Representative images of siRNA knockdowns described in panel A with GFP-Rab35 localization. L denotes lumen and arrowheads denote abnormal actin accumulations. White dotted lines mark sprout exterior. **d**–**g** Graphs of indicated sprout parameters across groups. *n* = number of sprouts. Error bars represent standard deviation, middle bars are the mean. **h** GFP-Rab35 localization in DENNd1a-c siRNA-treated sprouts. Localizations were binned to apical plasma membrane (PM, Rab35 > 80% at apical membrane), apical PM accumulations (non-continuous, visible puncta), equal PM (equally enriched at apical and basal membranes), and cytosolic. *n* = number of sprouts. NS = non-significant. Statistical significance was assessed with an unpaired t-test or a 1-way ANOVA followed by a Dunnett multiple comparisons test. Insets are areas of higher magnification. All experiments were done using human umbilical vein endothelial cells in triplicate.

localization. In 3D sprouts, inhibition of branching actin resulted in normal sprouting with elevated indices of actin puncta similar to the Rab35 knockdowns (Fig. 6b, Supplementary Fig. 10D; Supplementary Movies 4, 5). In 2D culture, CK-666 treatment rapidly depleted actin at the cell cortex (Fig. 6c). Rab35 prior to CK-666 administration exhibited a uniform distribution in the plasma membrane with enrichment at sites of actin. Following CK-666, Rab35 collapsed into discrete puncta scattered throughout the cytoplasm (Fig. 6c; Supplementary Movie 6). As a control, we performed the same experiment with Rab11a and did not observe any alteration in Rab11a localization with CK-666 treatment (Supplementary Fig. 10E). Using the same approach, we observed that DENNd1c was highly enriched at cortical actin and CK-666 effectively depleted DENNd1c from this actin population (Fig. 6d). Unlike Rab35, CK-666 treatment did not cause the formation of puncta, but the redistribution of DENNd1c to unaffected actin, such as F-actin towards the cell interior (Fig. 6d; Supplementary Movie 7). As a control, we treated cells with CK-666 expressing both Rab35 and Arp2. As expected, Arp2 was no longer located at the cell cortex, collapsing into puncta, while remaining adjacent to Rab35 (Fig. 6e; Supplementary Movie 8). These data suggest that Rab35 and DENNd1c are recruited to actin filaments.

To visualize Rab35's temporospatial recruitment to cortical actin, we employed a chemically switchable GFP-binding nanobody, termed ligand-modulated antibody fragments (LAMAs)[43]. This method sequesters GFP-tagged Rab35 at the mitochondria and then rapidly releases the protein upon drug treatment, enabling dynamic imaging of localization patterns (Supplementary Fig. 10F). Using GFP-Rab35, LAMA and TagRFP647-LifeAct[44] expressing cells, we released GFP-Rab35 from mitochondria and live-imaged its subsequent localization. Rab35 quickly localized to the cell periphery following trimethoprim (TMP) treatment, avoiding longer-lived F-actin (Fig. 6f and Supplementary Movie 9). When repeated with Arp2 and DENNd1c, Rab35 quickly (~2 min) localized to both proteins on the cell cortex (Fig. 6g, h; Supplementary Movies 10, 11). Next, we released Rab35, and then treated with CK-666 to determine how this association would be affected. Administration of CK-666 rapidly dissociated Rab35 and Arp2 at the cortex (Fig. 6i; Supplementary Movie 12). Lastly, to test if DENNd1c was responsible for recruiting Rab35 to branched actin, we knocked down DENNd1c and observed a significant reduction in Rab35's ability to localize to cortical Arp2 (Fig. 6j, k; Supplementary Movie 13). Overall, these data suggest that Rab35 is rapidly recruited to the actin cortex and is anchored by DENNd1c.

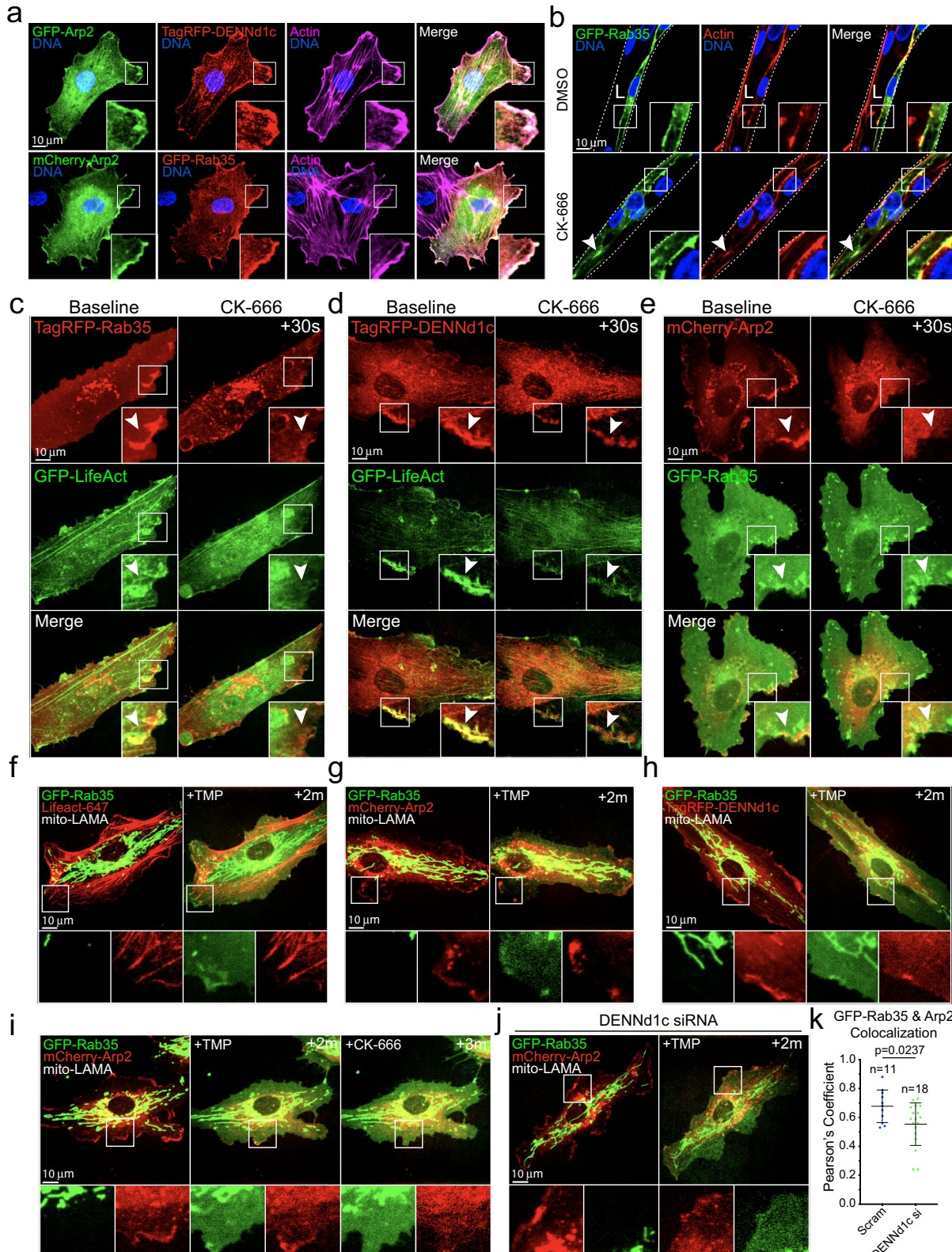

## Rab35 regulates actin assembly

Our next aim was to test whether Rab35 affected actin polymerization, per se. Prior literature indicates that Rab35 would increase actin polymerization via its purported trafficking interactions with Cdc42 and Rac1[37,38,40]; however, others have claimed Rab35 may act as a brake for actin polymerization through its association with MICAL-1[22]. To explore how Rab35 impacts actin in ECs, we transfected Rab35 variants

WT, CA, and DN into freely migrating ECs. It is well-established that lamellipodia protrusions and retractions are primarily mediated by local actin assembly and disassembly[45,46]. To monitor lamellipodia dynamics, we employed the open source software ADAPT[47]. Our analysis determined that only the Rab35-CA mutant significantly increased both the cells protrusive and retractive capabilities, a finding in line with enhanced migration[48] (Fig. 7a, b). Interestingly, knockdown of

**Fig. 6 | Rab35 localizes to cortical actin. a** Two-dimensional localization of GFP-Arp2 with DENNd1c (top panels) and GFP-Rab35 (bottom panels). **b** Representative images of DMSO and CK-666 (Arp Inhibitor) treated sprouts expressing GFP-Rab35. L denotes lumen. **c**, **d** Live imaging of GFP-Rab35 or TagRFP-DENNd1c with TagRFP647-LifeAct at baseline and after treatment with CK-666. White arrowheads denote the disappearance of Rab35 puncta over time. **e** Representative live-images of a cell expressing mCherry-Arp2 and GFP-Rab35 before and after CK-666 treatment. White arrowheads denote the disappearance of Rab35 puncta over time. **f** Live-image of a cell expressing GFP-Rab35, TagRFP647 (647)-LifeAct and ligand-modulated antibody fragments targeted to the mitochondria (mito-LAMA) before and after TMP administration. **g** Live-image of a cell expressing GFP-Rab35, mCherry-Arp2, and mito-LAMA before and after TMP administration. **h** Live-image of a cell expressing GFP-Rab35, TagRFP-DENNd1c, and mito-LAMA before and after TMP administration. **i** Live-image of a cell expressing GFP-Rab35, mCherry-Arp2, and mito-LAMA before and after TMP administration and then treated with CK-666. **j** Live-image of a cell expressing GFP-Rab35, mCherry-Arp2, and mito-LAMA treated with DENNd1c siRNA (si) before and after TMP administration. **k** Pearson's Coefficient of Rab35 and Arp between Scram and DENNd1c siRNA treated cells 2 min following TMP treatment. $n$ = number of cells. Error bars represent standard deviation, middle bars are the mean. Statistical significance was assessed with an unpaired t-test. Insets are areas of higher magnification. All experiments were done using human umbilical vein endothelial cells in triplicate.

Rab35 did not shift 2D-membrane dynamics significantly, potentially suggesting Rab35-based actin regulation may play a more central role in 3D sprouting.

We reasoned that if Rab35 was involved with actin polymerization, then knockdown of Rab35 would shift the balance between globular (G) and F-actin to skew more globular, assuming less filaments are being assembled. We isolated the G- and F-actin pools[49] and found that Rab35 knockdown significantly increased the G-actin ratio compared with control (Fig. 7c, d). In staining for G-actin using GC-globulin and phalloidin to detect F-actin[50], we observed an significant increase in the G/F-actin ratio in the absence of Rab35 as compared with controls (Fig. 7e, f). Co-staining for G/F actin while expressing Rab35, we validated Rab35 colocalized with both actin populations (Supplementary Fig. 10G). This data suggests that loss of Rab35 is associated with elevated cellular G-actin.

Using scanning electron microscopy, we visualized the filament network in ECs depleted of Rab35 or treated with CK-666. Qualitatively, there was reduced filament density in the lamellipodia regions of the Rab35 depleted and CK-666 treated conditions as compared with control (Fig. 7g). In Rab35 depleted ECs, we also observed elevated instances of disorganized actin bundles (Fig. 7g), potentially representing nodes of atypical actin polymerization. Overall, these results suggest that Rab35 is associated with regulating local sites of actin assembly.

### Loss of Rab35 promotes chronic cytoskeletal rearrangements

Cumulatively, our data suggests that Rab35 has a potent ability to perturb actin dynamics; however, we were still uncertain to how this was being achieved mechanistically. In other words, does the loss of Rab35 decrease actin polymerization programs or enhance it? Given Rab35 naturally localizes to the actin cortex in 2D, we investigated how loss of Rab35 impacted two actin networks interfacing through Rac1 or RhoA. Rac1 is a highly characterized GTPase that mediates branched actin polymerization, while RhoA mediates assembly of actin stress fibers[51]. To our surprise, we observed that both Rac1 and RhoA activity were elevated in the absence of Rab35 (Fig. 8a–d) This finding is in line with DENNd1c's ability to bind both actin populations; although, it is contrary to our previous assumption that loss of Rab35 reduced actin polymerization. To our knowledge, this is the first indication that loss of Rab35 promotes chronic actin rearrangements.

We next sought to reconcile our previous results showing elevated G-actin in Rab35 knockdown cells. Curious as to why stimulating two of the most prominent *pro*-actin polymerization pathways resulted in more G-actin, we treated cells with combinations of Rac1 and RhoA inhibitors (NSC23766 and Y27632) as well as activating compounds (CN02 and CN03). Over-activation of RhoA significantly skewed cells towards elevated G-actin, while over-activation of Rac1 did not affect G-/F-actin ratio (Fig. 8e–h). This result agrees with our finding that Rab35 loss of function biases the actin pool towards more globular state likely due to increased RhoA activity.

To understand how aberrant Rac1 and RhoA activation affected sprouting, we over-activated both Rac1 and RhoA in sprouts. Over-activation of either protein caused significantly increased lumen failures as well as generalized sprouting defects (Fig. 8i–l). To further confirm over-activation of RhoA downstream of Rab35 knockdown, we transduced sprouts with a RhoA localization sensor (C-terminus of Anillin)[52]. In control sprouts, ECs showed limited localization of the Biosensor on actin after lumenogenesis; however, in the Rab35 knockdown sprouts the biosensor was primarily distributed on F-actin (Supplementary Fig. 11A). Lastly, we used Rac1 and RhoA inhibitors to reduce chronic activation of these pathways on a Rab35 loss of function background. Indeed, dampening of Rac1, not RhoA, significantly improved the percentage of lumenized sprouts on a Rab35 knockdown background (Fig. 8m–o). Overall, this data suggests that loss of Rab35 promotes chronic over-activation of both Rac1 and RhoA; however, aberrant Rac1 activity is likely the underlying mechanism driving chronic actin remodeling and generalized sprout dysmorphogenesis.

### Rab35 is required for blood vessel development in zebrafish

We next generated a Rab35 knockout in zebrafish using CRISPR/Cas9 gene editing to test if Rab35 was also required for in vivo angiogenic processes[53]. In zebrafish, we targeted both Rab35 paralogs, Rab35A and Rab35B (Fig. 9a; Supplementary Fig. 11B). Double Rab35A/B knockout was embryonic lethal marked by a lack of normal development as compared with scramble guide injected controls, suggesting Rab35 is critical for normal embryonic development (Fig. 9b). However, we did produce a spectrum of developmental defects when the single-guide RNA amount was diluted. In a vascular Lifeact-GFP expressing line injected with a sublethal dosage of Rab35A/B single-guide RNA, we focused on actin defects. Here, we did not quantify vascular defects due to the generalized tissue dysmorphogenesis of these embryos; alternatively, our goal was to determine if similar actin accumulations occurred in vivo as observed in vitro. In line with our in vitro data, we observed a significant increase in actin aggregations in the Rab35A/B knockout group compared with controls (Fig. 9c, d). Vascular overexpression of the DN Rab35 mutant or treatment with CK-666 also promoted an increase in aberrant Rab35 accumulations, presumably bound to actin (Fig. 9e, f). To subvert the lethality of global Rab35A/B deletion, we generated chimeric embryos using blastomere transplants[54]. Transfer of Rab35A/B CRISPR injected cells into a WT host generated mosaic intersomitic blood vessels (ISVs) allowing for comparison of both WT and Rab35A/B null blood vessels side-by-side. Similar to in vitro results, Rab35A/B null ISVs were dysmorphic, marked by a thin appearance and the absence of a lumen as assessed by microangiography (Fig. 9g, h). Overall, these results indicate Rab35 is necessary for organismal viability and actin homeostasis in vivo.

## Discussion

The primary goal of this work was to interpret what of the many reported functions of Rab35 matters most during blood vessel morphogenesis by systematically characterizing Rab35 itself and the downstream effector pathways. Using a combination of 3D sprouting, biochemistry and in vivo gene editing, we demonstrate that Rab35's most prominent function is to regulate actin dynamics during angiogenesis. More specifically, we show that the GEF DENNd1c tethers active Rab35 to the actin cytoskeleton. Once localized to actin, Rab35

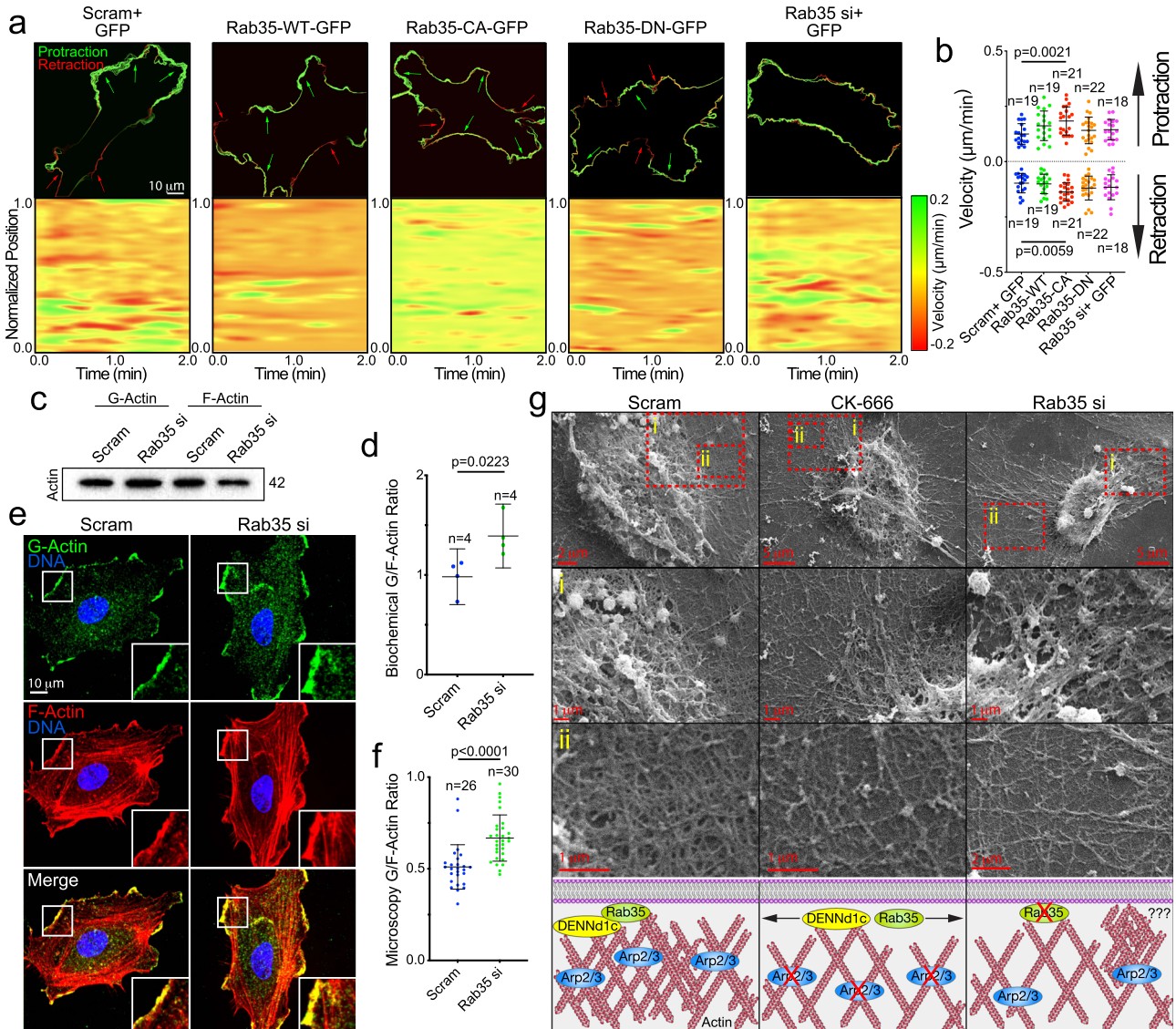

**Fig. 7 | Rab35 regulates actin dynamics. a** Top panels depict change in membrane velocities over time in described conditions. Green represents protraction and red represents retraction of cell membrane. Arrows indicate directionality. The bottom panels are heat maps in which the Red is indicative of retractive movement and green is protractive movement over time. Yellow indicates no change in velocity. **b** Quantification of cell membrane velocities between indicated groups. Above the dashed line is the protractive velocities and below the dashed line is retractive velocities. *n* = number of cells. Error bars represent standard deviation, middle bars are the mean. **c** Western blot of globular (*G*) and filamentous (*F*) actin in siRNA (si)-treated groups. **d** Quantification of the ratio of globular to filamentous actin from blots represented in panel (**c**). *n* = number of cells. Error bars represent standard deviation, middle bars are the mean. **e** Representative images of cells stained for globular and filamentous actin between indicated conditions. **f** Quantification of the ratio of globular to filamentous actin fluorescent intensities. *n* = number of cells. Error bars represent standard deviation, middle bars are the mean. **g** Scanning electron microscopy of filament network between groups. Top panel is the lowest magnification with higher magnifications in panels (i) and (ii). Bottom- cartoon representation of SEM filament network and hypothesized role of Rab35. NS = non-significant. Statistical significance was assessed with an unpaired t-test or a 1-way ANOVA followed by a Dunnett multiple comparisons test. Insets are areas of higher magnification. All experiments were done using human umbilical vein endothelial cells in triplicate.

limits actin polymerization and remodeling required for sprout formation (Fig. 8o). To our knowledge, this is the only investigation demonstrating the requirement of Rab35 for blood vessel function and the only investigation in any tissue dissecting Rab35's most dominant biological role accounting for the most prominent effector pathways.

The current project was originally aimed to characterize how podocalyxin was trafficking in ECs, as this is still an outstanding question in the field. Our past work demonstrated Rab27a, that was largely implicated in podocalyxin trafficking in epithelial cells, was not related to this pathway[7], thus our very next candidate was Rab35. Others have comprehensively established a direct association between Rab35 and podocalyxin as well as the downstream impact on lumen biogenesis[16,17]. Our data in the current investigation once again shows

that endothelial trafficking signatures greatly differ from epithelial programs. We expansively tested for both localization and direct binding interaction between Rab35 and podocalyxin of which we found none. Although, this negative result prompted us to further investigate Rab35 function during angiogenic sprouting.

Rab35 has been shown to have many roles that vary by tissue type, organism, and developmental stage. In distilling the literature, it can be argued that Rab35 has six major effectors that mediate its function in vertebrates: Rusc2, MICAL-L1, MICAL1, ACAP2, OCRL, and Fascin. We began by first establishing that Rab35 was required for sprouting, and then determined how each effector contributed to the loss of Rab35 phenotype. Surprisingly, most effectors failed to directly bind Rab35. ACAP2 exhibited the best phenotype for

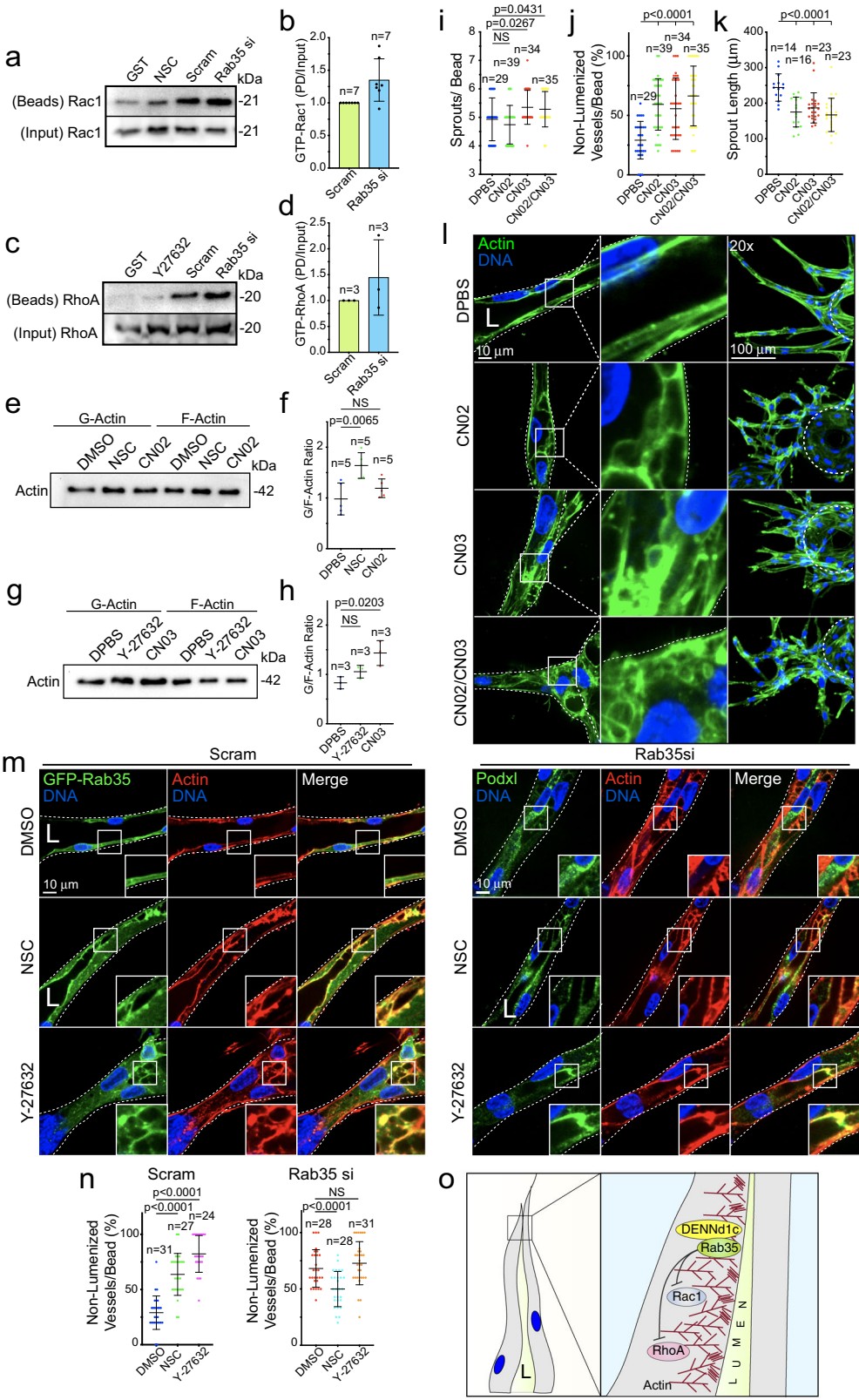

recapitulating the Rab35 loss of function effect. The predominant hypothesis is that GTP Rab35 binds ACAP2 sequestering its ability to inactivate Arf6, resulting in a gain of function for Arf6. In ECs, we confirmed direct binding between ACAP2, and also observed that Rab35 knockdown increased Arf6 activity as previously reported[18–20,29,32,33,55,56]. However, overexpression of CA Arf6 did not phenocopy the Rab35 loss of function effect on sprouting,

suggesting that aberrantly high Arf6 activity was likely not causing sprout dysmorphogenesis in the absence of Rab35.

A major finding was that the GEF DENNd1c played a key role in Rab35 function. Canonically, GEFs primarily convert proteins from a GDP to GTP-bound state; however, DENNd1c is evolutionarily divergent from both DENNd1a/b that solely control Rab35 GTPase activity[38]. In our hands, DENNd1c localized Rab35 to actin fibrils. Knockdown of

**Fig. 8 | Loss of Rab35 promotes chronic actin remodeling. a** Representative western blot pulldown for activated Rac1. Cells were treated with scramble (Scram) and Rab35 siRNA (si) or treated with Rac Inhibitor NSC23766 (NSC) then probed for active (GTP) Rac1. **b** Quantification of Rac1 activity between indicated groups normalized to control. *n* = number of pull-downs. Error bars represent standard deviation, middle bars are the mean. **c** Representative western blot pulldown for activated RhoA. Cells were treated with scramble and Rab35 siRNA or ROCK inhibitor (Y27632). **d** Quantification of RhoA activity between indicated groups normalized to control. *n* = number of pull-downs. Error bars represent standard deviation, middle bars are the mean. **e** Western blot of globular (*G*) and filamentous (*F*) actin populations treated with DPBS (vehicle), NSC, or Rac1 activator (CN02). **f** Quantification of the ratio of globular to filamentous actin from blots represented in panel (**e**). *n* = number of blots. Error bars represent standard deviation, middle bars are the mean. **g** Western blot of globular and filamentous actin population treated with DPBS, Y27632, and RhoA activator (CN03). **h** Quantification of the ratio of globular to filamentous actin from blots represented in panel (**g**). *n* = number of blots. Error bars represent standard deviation, middle bars are the mean. **i–k** Graphs of indicated sprout parameters across groups. *n* = number of sprouts. Error bars represent standard deviation, middle bars are the mean. **l** Representative images of sprouts treated with indicated compounds. **m** Representative images of Scram (left figure) and Rab35 si (right figure) treated sprouts with indicated drug treatments. **n** Quantification of non-lumenized vessels/bead from experiments in panel (**m**). *n* = number of sprouts. Error bars represent standard deviation, middle bars are the mean. **o** Model of Rab35 mechanism. In all panels, L denotes lumen. NS = non-significant. Statistical significance was assessed with an unpaired t-test or a 1-way ANOVA followed by a Dunnett multiple comparisons test. Insets are areas of higher magnification. All experiments were done using human umbilical vein endothelial cells in triplicate.

DENNd1c strongly phenocopied loss of Rab35 suggesting that localization to actin is a primary function of Rab35 during sprouting angiogenesis.

Actin plays a pivotal role in angiogenesis both from a cell migration and vessel stabilization aspect[57–62]. Loss of normal actin architecture has been shown to drastically affect virtually all facets of blood vessel formation[63–67]. In this sense, our results are not surprising in that actin misregulation promoted such a profound negative effect on sprouting parameters. However, given Rab35's broad scope of function as well as never being characterized in angiogenic processes, it would be exceedingly hard to predict. Our results paint a scenario that trafficking-based regulators can control vital crosstalk with the actin cytoskeleton. Our results strongly indicate that Rab35 limits the activation of Rac1 and RhoA-mediated cytoskeletal rearrangement. Whereby loss of Rab35 creates a scenario in which the actin cytoskeleton is never fully stabilized promoting gross polarity defects in sprouts. It is still an outstanding question as to how this activation is achieved or what is the Rab35 interactome. Importantly, our results may provide an additional mechanism of why loss of Rab35 causes elevated cancer invasiveness through loss of tissue architecture[68–70].

Overall, our investigation is the first to systematically rule out other known Rab35 pathways, highlighting Rab35's function in mediating actin dynamics during blood vessel formation in vitro and in vivo. In general, we contend that mapping endothelial trafficking patterns will shed important light on how ECs orchestrate blood vessel formation by integrating both cell-autonomous and collective-cell signaling.

## Methods

All research complied with the University of Denver Institutional Biosafety Committee (IBC) and Institutional Animal Care and Use Committee (IACUC).

### Reagents

All reagents, siRNA, and plasmid information are listed in the reagents table in the supplementary information (Supplementary Tables 1–5).

### Cell culture

Pooled Human umbilical vein endothelial cells (HUVECs) were purchased from PromoCell and cultured in proprietary media for 2–5 passages. All cells were maintained in a humidified incubator at 37 °C and 5% $CO_2$. Small interfering RNA was introduced into primary HUVEC using the Neon® transfection system (ThermoFisher) resuspended to a 20 μM stock concentration and used at 0.5 μM. Normal human lung fibroblasts and HEK-A were maintained in Dulbeccos Modified Medium (DMEM) supplemented with 10% fetal bovine serum and antibiotics. For two-dimensional live-imaging experiments, cells were imaged for one minute at baseline before treatment with CK-666 (1 μM), and then imaged for an additional two minutes using 5 s intervals. For ligand-modulated antibody fragments tether to the mitochondria (Mito-

LAMA) experiments procedures were carried out as previously described[43]. All plasmids are listed in the Supplementary Table 4.

### Sprouting angiogenesis assay

Fibrin-bead assay was performed as reported by Nakatsu et al. 2007[26]. Briefly, HUVECs were coated onto microcarrier beads (Amersham) and plated overnight. SiRNA-treatment or viral transduction was performed the same day the beads were coated (Supplementary Table 1). Endothelial cell covered microbeads were embedded in a fibrin matrix. Once the clot was formed media was overlaid along with 100,000 NHLFs. Media was changed daily along with monitoring of sprout development. Sprout characteristics were quantified in the following manner. Sprout numbers were determined by counting the number of multicellular sprouts (sprouts that did not contain at least 3 cells were not used in the analysis) emanating from an individual microcarrier beads across multiple beads in each experiment. Sprout lengths were determined by measuring the length of a multicellular sprout beginning from the tip of the sprout to the microcarrier bead surface across multiple beads. Percent of non-lumenized sprouts were determined by quantifying the proportion of multicellular sprouts whose length (microcarrier bead surface to sprout tip) was less than 80% lumenized across multiple beads. Sprout widths were determined by measuring the sprout width at the midpoint between the tip and the microcarrier bead across multiple beads. Actin accumulations were defined by actin puncta with a diameter greater than 1.5 μm. Experimental repeats are defined as an independent experiment in which multiple cultures, containing numerous sprouting beads were quantified; this process of quantifying multiple parameters across many beads and several cultures was replicated on different days for each experimental repeat.

### Lentivirus and adenovirus generation and transduction

Lentivirus was generated by using the LR Gateway Cloning method[25]. For lentiviral generation genes of interest and fluorescent proteins were isolated and incorporated into a pME backbone via Gibson reaction[71]. Following confirmation of the plasmid by sequencing the pME entry plasmid was mixed with the destination vector and LR Clonase. The destination vector used in this study was pLenti CMV Neo DEST (705-1) (gift from Eric Campeau & Paul Kaufman; Addgene plasmid #17392). Once validated, the destination plasmids were transfected with the three required viral protein plasmids: pMDLg/pRRE (gift from Didier Trono; Addgene plasmid # 12251), pVSVG (gift from Bob Weinberg; Addgene plasmid #8454) and psPAX2 (gift from Didier Trono; Addgene plasmid #12260) into HEK 293 cells. The transfected HEKs had media changed 4 h post transfection. Transfected cells incubated for 3 days and virus was harvested.

Adenoviral constructs and viral particles were created using the Adeasy viral cloning protocol[72]. Briefly, transgenes were cloned into a pShuttle-CMV plasmid (gift from Bert Vogelstein; Addgene plasmid #16403) via Gibson Assembly. PShuttle-CMV plasmids were then digested overnight with MssI (ThermoFisher) and Linearized pShuttle-

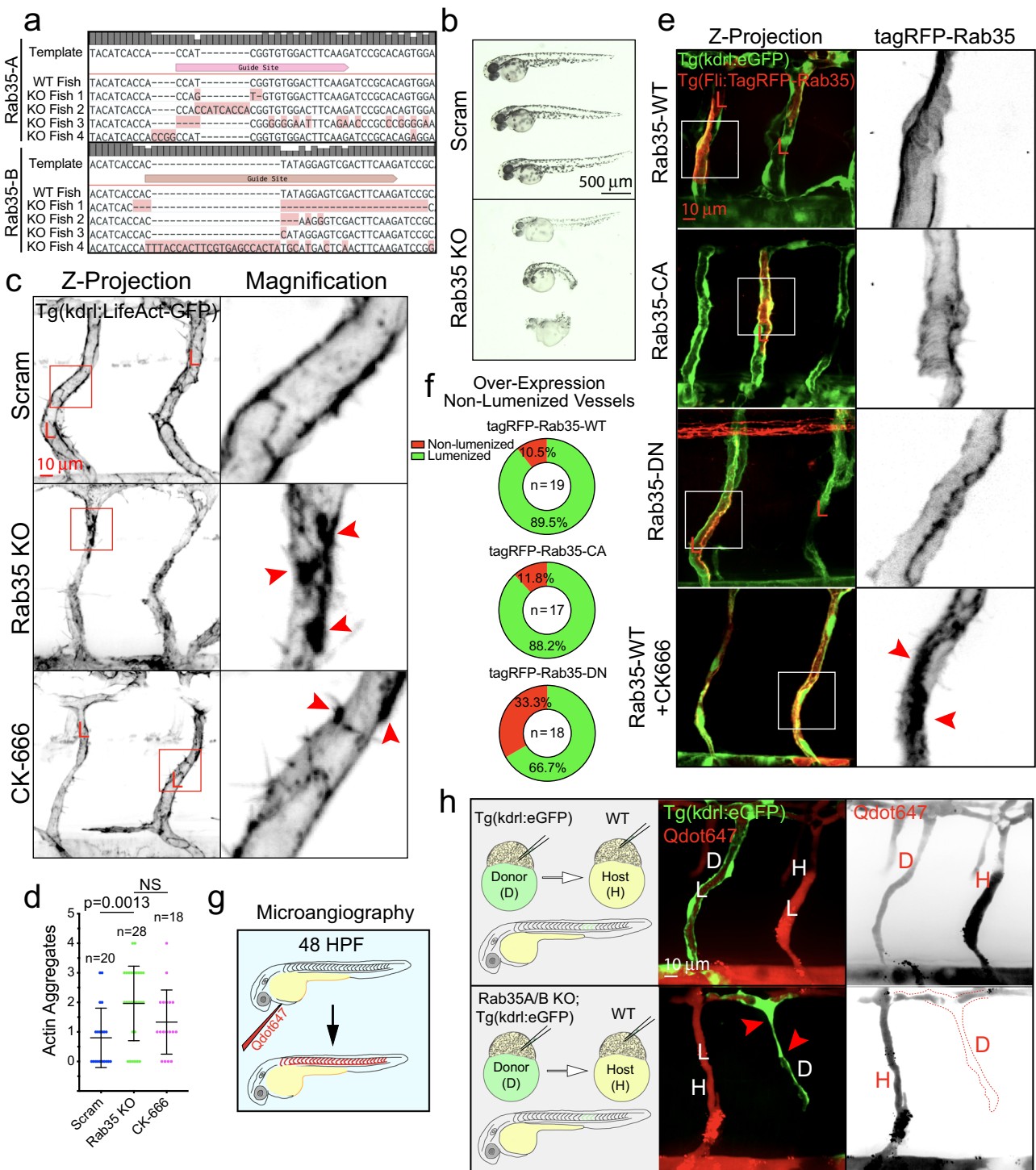

**Fig. 9 | Rab35 is required for blood vessel development in zebrafish. a** CRISPR-mediated knockout of Rab35A/B confirmation by sequencing. Four random fish were sequenced following CRISPR/guide injections. **b** Zebrafish morphology at 48 h post fertilization (hpf) post injection of scramble (Scram) and Rab35A/B CRISPR guides. **c** Representative images of intersomitic blood vessels (ISVs) of Scram and Rab35A/B knockout, as well as CK-666 (Arp Inhibitor), treated zebrafish at 48 hpf expressing endothelial specific LifeAct-GFP. Red arrowheads indicate abnormal aggregates of actin. **d** Quantification of actin aggregates between groups. *n* = number of ISVs. A minimum of 5 fish were used per group. Error bars represent standard deviation, middle bars are the mean. **e** Representative images of mosaic expression of Tag-RFP-Rab35 WT (top row), CA (second row), DN (third row), and WT with CK-666 treatment in zebrafish at 48 hpf. Red arrowheads depict excess of

Rab35 at the plasma membrane. **f** Percentage of non-lumenized vessels at 48 hpf between groups mentioned in panel (**e**). *n* = number of ISVs. **g** Cartoon representation of microangiography in zebrafish larvae using quantum dots 647 (Qdot647) at 48 hpf. **h** Representative images of ISVs after transplantation of Tg(kdrl:GFP) donor (D) into Tg(kdrl:mCherry) host (H) (top panels). Bottom panels-representative images of ISVs after transplantation of Rab35A/B knockout donor cells from Tg(kdrl:GFP) line into Tg(kdrl:mCherry) host. Red arrowheads indicate lumen failure. NS = non-significant. Error bars represent standard deviation. Statistical significance was assessed with an unpaired t-test or a 1-way ANOVA followed by a Dunnett multiple comparisons test. Insets are areas of higher magnification. All experiments were done in triplicate.

CMV plasmids were transformed into the final viral backbone using electrocompetent AdEasier-1 cells (gift from Bert Vogelstein; Addgene, #16399). Successful incorporation of pShuttle-CMV construct into AdEasier-1 cells confirmed via digestion with PacI (ThermoFisher). 5000 ng plasmid was then digested at 37 °C overnight, then 85 °C for 10 min, and transfected in a 3:1 polyethylenimine (PEI, Sigma):DNA ratio into 70% confluent HEK 293A cells (ThermoFisher) in a T-25 flask.

Over the course of 2–4 weeks, fluorescent cells became swollen and budded off the plate. Once approximately 70% of the cells had lifted off the plate, cells were scraped off and spun down at 2000 rpm for 5 min in a 15 mL conical tube. The supernatant was aspirated, and cells were resuspended in 1 mL PBS. Cells were then lysed by 3 consecutive quick freeze-thaw cycles in liquid nitrogen, spun down for 5 min at 2000 rpm, and supernatant was added to 2qty 70% confluent T-75 flasks. Propagation continued and collection repeated for infection of 10–15 cm dishes. After collection and 4 freeze thaw cycles of virus collected from 10 to 15 cm dishes, 8 mL viral supernatant was collected and combined with 4.4 g CsCl (Sigma) in 10 mL PBS. Solution was overlaid with mineral oil and spun at 32,000 rpm at 10 °C for 18 h. Viral fraction was collected with a syringe and stored in a 1:1 ratio with a storage buffer containing 10 mM Tris, pH 8.0, 100 mM NaCl, 0.1 percent BSA, and 50% glycerol. HUVEC were treated with virus for 16 h at a 1/10,000 final dilution in all cell culture experiments.

## Antibody feeding assay

Antibody feeding assay was carried out as previously described[73]. Briefly, cells were moved to 4 °C for 30 min, and then β1-integrin antibody was added to the culture for an additional 30 min. Following incubation, cells were washed and moved back into the 37 °C incubator for 20 min and then fixed with 4% PFA. β1-integrin antibody was added once more for 45 min to label extracellular integrins, washed, and then incubated with the secondary antibody. The secondary was washed, and cells were permeabilized with 0.5% Triton-X for 10 min to gain access to the endocytosed β1-integrin pool. Then a complementary secondary antibody was added for 20 min to label the endocytosed integrins.

## Wound healing assay

Treated cells were moved to Ibidi culture insert plates with a two well silicone insert allowing for a defined cell-free gap. At 3 days post siRNA treatment the silicone insert was removed, and cells were allowed to migrate for 6 h. Thereafter, cells were fixed, and immunohistochemistry was performed. The distance traveled into the cell free space was measured between groups.

## Immunoblotting and protein pull-down

HUVEC cultures were trypsinized and lysed using Ripa buffer containing 1× ProBlock™ Protease Inhibitor Cocktail-50 (GoldBio) and processed as previously described[7]. Rac1 and RhoA activity blots were performed using commercially available kits (Cytoskeleton; Supplementary Table 1).

## Detection of globular and filamentous actin

Globular and filamentous actin ratios were determined by western blot as described by commercially available G-actin/ F-actin In Vivo Assay Kit (Supplemental Table 1). Globular and filamentous immunocytochemistry was performed as previously described[50]. Briefly, cells were fixed with 4% PFA for 10 min and permeabilized in ice cold acetone for 5 min and washed. Cells were then incubated for 15 min in 2% BSA with globular actin-binding protein GC globulin (Sigma). Following incubation, cells were washed three times in PBS. After washes cells incubated with an anti-GC antibody in BSA for 15 min, washed three times, and incubated in anti-rabbit-555 secondary prior to imaging.

## Tracking of cell dynamics

Cell tracking was performed using ADAPT as previously described[47].

## Immunofluorescence and microscopy

For immunofluorescence imaging, HUVECs were fixed with 4% paraformaldehyde (PFA) for 7 min. ECs were then washed three times with PBS and permeabilized with 0.5% Triton-X (Sigma) for 10 min. After permeabilization, cells were washed three times with PBS. ECs were then blocked with 2% bovine serum albumin (BSA) for 30 min. Once blocked, primary antibodies were incubated for approximately 4–24 h. Thereafter, primary antibodies were removed, and the cells were washed 3 times with PBS. Secondary antibody with 2% BSA were added and incubated for approximately 1–2 h, washed 3 times with PBS, and mounted on a slide for imaging. All primary and secondary antibodies are listed in the Supplemental Data 3. All images were taken on a Nikon Eclipse Ti inverted microscope equipped with a CSU-X1 Yokogawa spinning disk field scanning confocal system and a Hamamatusu EM-CCD digital camera. Images were captured using a Nikon Plan Apo 60x NA 1.40 oil objective using Olympus type F immersion oil NA 1.518, Nikon Apo LWD 20× NA 0.95 or Nikon Apo LWD 40× NA 1.15 water objective. All images were processed using ImageJ (FIJI).

## Zebrafish transplantation, microangiography, and gene editing

All animal studies were approved by the University of Denver IACUC in accordance with AAALAC recommendations. Zebrafish (Danio rerio, AB strain) larvae between 24 and 72 h post fertilization were used for all animal experiments, no adults were used. Zebrafish transplantations were performed as previously described[74]. Briefly, cells were harvested at the blastula stage from a tg(kdrl:mCherry) line and treated with CRISPR (described below) line using an Eppendorf CellTram and deposited into recipients harboring a tg(kdrl:eGFP) transgene allowing us to distinguish between host and recipient blood vessels.

For microangiography 48 hpf embryos were (anesthetized) with 1X tricaine for approximately 20 min prior to perfusion. Embryos were then loaded ventral side up onto an injection agarose facing the injection needle. Qdots (ThermoFisher) were sonicated prior to injection. Qdots were loaded into a pulled capillary needle connected to an Eppendorf CellTram and 1–3 µl of perfusion solution was injected into the pericardial cavity. Once successfully perfused, embryos were embedded in 0.7% low melt agarose and imaged promptly. Images were taken on a Nikon Eclipse Ti inverted microscope equipped with a CSU-X1 Yokogawa spinning disk field scanning confocal system and a Hamamatusu EM-CCD digital camera using either Nikon Apo LWD 20× NA 0.95 or Nikon Apo LWD 40× NA 1.15 water objective.

Tol2-mediated transgenesis was used to generate mosaic intersomitic blood vessels as previously described[2,3]. Briefly, Tol2 transposase mRNA were synthesized (pT3TS-Tol2 was a gift from Stephen Ekker, Addgene plasmid # 31831)[4] using an SP6 RNA polymerase (mMessage Machine, ThermoFisher). A total of 400 ng of transposase and 200 ng of plasmid vector were combined and brought up to 10 µL with phenol red in ddH2O. The mixture was injected into embryos at the 1–2 cell stage. Injected zebrafish were screened for mosaic expression at 48 hpf and imaged.

CRISPR/cas9-mediated knockouts were performed as previously described[5]. Briefly, equal volumes of chemically synthesized AltR® crRNA (100 µM) and tracrRNAr RNA (100 µM) were annealed by heating and gradual cooling to room temperature. Thereafter the 50:50 crRNA:tracrRNA duplex stock solution was further diluted to 25 µM using supplied duplex buffer. Prior to injection 25 µM crRNA:tracrRNA duplex stock solution was mixed with 25 µM Cas9 protein (Alt-R® S.p. Cas9 nuclease, v.3, IDT) stock solution in 20 mM HEPES-NaOH (pH 7.5), 350 mM KCl, 20% glycerol) and diluted to 5 µM by diluting with water. Prior to microinjection, the RNP complex solution was incubated at 37 °C, 5 min and then placed on ice. The injection mixture was micro-injected into 1–2 cell stage embryos. Crispant DNA

was retrieved via PCR and subjected to sanger sequencing to visualize indel formation (Supplementary Table 5).

## Zebrafish live imaging and quantification

All zebrafish presented were imaged at 48 hpf. Prior to imaging, embryos were treated with 1% Tricaine for 20 min and afterwards embedded in 0.7% low melt agarose. Live imaging of Zebrafish intersomic vessels (ISVs) were performed using the spinning-disk confocal microscopy system mentioned above. ISVs that were analyzed were between the end of the yolk extension and tail. Parameters measured included ISV number, number of non-lumenized vessels (no visible separation between opposing endothelial cells in ISVs), and number of actin accumulations (actin accumulations with a diameter greater than 4 μm).

## Scanning electron microscopy

Cells fixed for SEM were followed the procedure outlined by Watanabe, et al.[75]. Scanning electron microscopy was performed at the University of Colorado Anschutz Medical Campus by Dr. Eric Wortchow.

## Statistical analysis

Experiments were repeated a minimum of three times. Statistical analysis and graphing were performed using GraphPad Prism. Statistical significance was assessed with a student's unpaired t-test for a two-group comparison. Multiple group comparisons were carried out using a one-way analysis of variance (ANOVA) followed by a Dunnett multiple comparisons test. Data was scrutinized for normality using Kolmogorov–Smirnov test. Zebrafish sex distribution was not adjusted as sex determination did not occur at the stage of development in which the specimens were assayed. Statistical significance set a priori at $p < 0.05$.

## Reporting summary

Further information on research design is available in the Nature Research Reporting Summary linked to this article.

# Data availability

Numeric data for this study can be found in the Supplementary Data File. The authors will make any other data, analytic methods, and study materials available to other researchers upon written request. Source data are provided with this paper.

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

## Acknowledgements

Work was supported by funding from the National Heart Lung Blood Institute (Grant 1R56HL148450-01 (E.J.K.), R15HL156106-01A1 (E.J.K.), R01HL155921-01A1) (E.J.K.).

## Author contributions

C.R.F., H.K., and E.J.K. performed all experiments. C.R.F. and E.J.K. wrote the manuscript.

## Competing interests

The authors declare no competing interests.
