## [Peer Review File · Nature Communications]

Rab35 Governs Apicobasal Polarity Through Regulation of Actin Dynamics During Sprouting AngiogenesisREVIEWER COMMENTS

Reviewer #1 (Remarks to the Author):

In this manuscript, Kushner's group investigated the possible role of Rab35 GTPase in blood vessel development. They found that depletion of Rab35 in HUVECs and in zebrafish resulted in reduced sprout formation from fibrin-beads and abnormal vessel development, respectively. They also showed that Rab35 regulates actin assembly required for proper angiogenesis, possibly through interaction with an unidentified Rab35 effector. While the involvement of Rab35 in blood vessel development is interesting, the present manuscript completely lacks mechanistic insights into how Rab35 regulates actin assembly during angiogenesis. Since it has already been reported that Rab35 regulates formation of cellular protrusions, such as seamless tubes in fruit flies (Nat. Cell Biol. 2012;14:386-393), cilia (EMBO Rep. 2019;20:e47625), and neurites, apicobasal polarity in epithelial cells, and actin dynamics in several cell types, the present findings themselves are not so surprising. "Identification and characterization of a novel Rab35 effector(s)" that functions together with Rab35 in blood vessel development is necessary before publication.

Other specific points:

1. Although the authors stated in the text that they analyzed known Rab35 effectors by a holistic approach, they tested only three Rab35 effectors, ACAP2, OCRL, and MICAL-L1. Two other important actin-regulating Rab35 effectors, MICAL1 (Nat. Commun. 2017;8:14528) and Fascin (Science 2009;325:1250-1254), are missing in this manuscript. The authors should test their involvement in angiogenesis.
2. Important control experiments are missing in Fig. 2C. The authors should also test Scram+Rab35-CA and Scram+Rab35-DN.
3. Knockdown of Rab35 effectors shown in Fig. 3B is not so efficient, in comparison with the Rab35 knockdown (Fig. 1C). This raises the possibility that insufficient knockdown of Rab35 effectors in HUVECs would mask phenotypes in angiogenesis. The authors need to evaluate the knockdown phenotypes by using more effective siRNAs. Also, knockdown efficiencies in Fig. 1C, Fig. 3B, Fig. 4B, and Fig. S4F should be calculated.
4. Based on the results shown in Fig. S7D and E, the authors suggested that ACAP2 does not interact with or act on Arf6. However, ACAP2 is an enzyme (Arf6-GTPase activating protein) and does not need to stably interact or colocalize with its substrate. The authors should consider this point.
5. Arf6 overexpression phenotypes (Fig. S7H) seem to be opposite to Rab35 overexpression phenotypes (Fig. 2C). Rab35-DN increased non-lumenized sprouts, whereas Arf6-DN decreased them, suggesting that both Rab35 and Arf6 are involved in blood vessel development. Although the authors claimed that the level of active Arf6 was not changed even after Rab35 depletion or overexpression without showing quantitative data, this reviewer thinks that inactivation of Arf6 would occur locally (e.g., at the apical membrane alone). To address this issue, effect of plasma membrane-targeted ACAP2 on angiogenesis should be tested.
6. In Fig. 4B, knockdown efficiency of DENNd1a-c should be investigated by Western blotting, the same as in Fig. S8B. Mr size of DENNd1a-c should also be provided in Fig. S8B.
7. Other minor points:
 - The manuscript contains a lot of typos. The authors should carefully check the manuscript. e.g., drosophila  Drosophila (line 71), 37°C degree  37°C (line 218), Laria-Bertani  Luria-Bertani (line 246), E Coli  E. coli (line 247), MICAL-L1  MICAL1 (line 561), and RUSC  RUSC2 (lines 412, 641 and 644).
 - Information about DENNd1a-c antibodies is missing in Resource Table.

Reviewer #2 (Remarks to the Author):

This interesting manuscript by Francis et. al explores the unexplored roles of Rab35 in sprouting angiogenesis. The authors apply a combination of in vitro and in vivo tools to suggest how this small GTPase modulates cytoskeletal events underpinning vessel formation. In this context, the authors argue that Rab35 plays its most critical role in apicobasal polarity with severe implications in lumenogenesis. They show that the GEF DENNd1c does not activate Rab35 in endothelial cells, but recruits it to sites of actin polymerization. This is a very elegant study with robust data and sound experimental controls. Overall the manuscript is very well presented, clear and well written. Nevertheless, an overall sense of lack of novelty runs across the manuscript. Although the authors claim "a novel scenario that trafficking-based regulators can control vital crosstalk with the actin cytoskeleton", this has been shown elsewhere. For example, the McPherson and Scott labs have linked RAB35 to Fascin, whilst Sasaki and colleagues have done extensive work to dissect the links between RAB13 and Filamin activity. Furthermore, several points could be addressed to make the current findings more compelling.

Major comments:

1. The role of Rab35 in cell migration is widely overlooked. Rab35 loss-of-function in the fibrin assays shows sprouts that "appeared stubby", are shorter and less abundant than in control cultures. (Figure 1D-F). Given the cell motility roles attributed to Rab35 in other systems, it is likely that Rab35 modulates endothelial migration steps preceding lumen formation, explaining the aforementioned loss-of-function phenotypes. The authors then claim that no cell motility defects are observed in Fig. S5A, but they do not carry out a comprehensive analysis of motility parameters. Of note, these experiments are performed in 2D and some caution must be taken when extrapolating to 3D scenarios.
2. If indeed Rab35 only plays critical roles in endothelial cell apicobasal polarity (to control lumen formation and not exerting effects in endothelial migration), did the authors explore if loss of Rab35 affects polarity in quiescent endothelial cells prior to sprouting? Could the perturbed apicobasal polarity impair cytoskeleton arrangements during tip cell selection?
3. In line with the prior comments, it is relevant to be clear about the role of Rab35 in tip cells. The authors disregard the topic mentioning that Rab35 "had no preference for tip cell" positioning. It is not very clear how they infer this without 3D imaging.
4. The authors demonstrate that RAB35 does not act via the ACAP2-Arf6 axis, but do not look for other potential effectors in endothelial cells that could explain the modulation of actin. Could Rab35 regulate Fascin? The material and methods section indicates the use of a Fascin inhibitor and Fascin plasmid but to my knowledge, no data in the manuscript reflects this.
5. The in vivo work is interesting and very relevant. Although the vital role played by Rab35 may explain the major developmental defects observed in CRISPR mutants, it is critical to demonstrate that the CRISPR guides do not have potential off-targets.
6. The use of a Rab35 biosensor to prove exactly where this small GTPase is active during sprouting angiogenesis would significantly substantiate the findings.

Minor comments:

1. Because siRNAs drop efficiency over time and thus, it would be useful to indicate the stages of sprout analysis and Western blotting. It is not clear from the methods section how long the fibrin sprouts are cultured or when the siRNA efficiencies are confirmed.
2. Why did the authors present error bars in the form of 95% confidence intervals? Although valid, this is harder to interpret than standard deviation.
3. In line 367 the authors mention that Rab35 is necessary for sprout function. I don't think that this is proven at this stage (although this is later on shown with in vivo data).
4. In Figure 3B, the siRNA knock down of Rab35 effectors is not very efficient. Could loss of a small amount of protein justify the phenotypes? Could the authors try other siRNAs?
5. The authors state "Arp2 was no longer located on actin" in line 530, but there is no actin

marker/staining in Figure 5E.

6. The authors mention that Rab35-CA alters protrusions formation and retraction “in line with enhanced migration”. Where is enhanced migration shown?

7. Some plasmids are missing in material and methods.

OVERVIEW OF CHANGES. We very much appreciate the reviewers considerable time and effort in critiquing our manuscript. As both reviewers stated, we agree that our original findings did not highlight a singular mechanism for how Rab35 directly modulates actin and downstream sprout polarity. We would contend that literature in other cell types referenced by Reviewer 1 and 2 provides no direct mechanistic evidence into how Rab35 alters actin signaling in mammalian tissue; thus, this remains a largely unresolved question in the field, and completely unexplored in blood vessel development. To this point, we are very grateful for the reviewer's push for further mechanistic exploration as we believe we have isolated a signaling network in ECs in attempting to satisfy this criticism. Our new results shed light on some of the controversial signaling, such as Rab35's role in spindle formation, or impact on microtubules reported by other investigations as our newly identified actin regulators can effect these pathways. In general, our new results significantly increase our understanding of this trafficking network in ECs and other tissues.

It has been inferred that Rab35 activates Rac1 and actin polymerization in epithelial systems or in non-mammalian cells[1]. To contextualize this signaling connection, the bulk of the evidence for an interplay between Rab35 with Rac1 was primarily inferred through colocalization studies [2, 3]. To our knowledge, no other reports detail a concrete signaling mechanism bridging the Rab35 phenotype in vertebrate animals or associated 3D organoids to proteins that directly influence actin. Several papers do show Rab35 alters polarity signaling; however, these reports do not add a mechanistic explanation for this phenotype. To provide a mechanism for our observations, we focused on Rab35's localization at the plasma membrane and primary phenotype in altering actin by further exploring both Rac1 and RhoA GTPases. Previous literature would indicate that loss of Rab35 would decrease Rac1; however, to our surprise, knockdown of Rab35 increased both Rac1 and RhoA activity in ECs. This finding was contrary to our original thinking that loss of Rab35 decreased actin polymerization, supported by cellular shifts in globular/filamentous actin. Unexpectedly, we discovered that chronic overactivation of RhoA, in particular, significantly increased the cellular globular actin pool through unknown mechanisms, validating our previous observations. Lastly, we demonstrate that chronic Rac1-mediated actin remodeling is the underlying issue promoting dysmorphic sprouting when Rab35 is ablated.

This paradigm of upregulation of Rac1 and RhoA in the absence of Rab35 has not been previously described, nor tested as comprehensively as we have laid it out in this report. Importantly, this finding is very much in-line with Rab35's association with cancer progression, our results provide a strong mechanistic basis for the loss of tissue architecture in tumorigenesis. Overall, our aim was to thoroughly characterize how Rab35 participates in angiogenic processes, the revised manuscript contains substantially more data to better solidify the requested mechanism and to remedy other minor criticisms. Please see our individual responses below.

REVIEWER COMMENTS

Reviewer #1 (Remarks to the Author):

In this manuscript, Kushner's group investigated the possible role of Rab35 GTPase in blood vessel development. They found that depletion of Rab35 in HUVECs and in zebrafish resulted in reduced sprout formation from fibrin-beads and abnormal vessel development, respectively. They also showed that Rab35 regulates actin assembly required for proper angiogenesis, possibly through interaction with an unidentified Rab35 effector. While the involvement of Rab35 in blood vessel development is interesting, the present manuscript completely lacks mechanistic insights into how Rab35 regulates actin assembly during angiogenesis. Since it has already been reported that Rab35 regulates formation of cellular protrusions,

such as seamless tubes in fruit flies (Nat. Cell Biol. 2012;14:386-393), cilia (EMBO Rep. 2019;20:e47625), and neurites, apicobasal polarity in epithelial cells, and actin dynamics in several cell types, the present findings themselves are not so surprising. "Identification and characterization of a novel Rab35 effector(s)" that functions together with Rab35 in blood vessel development is necessary before publication.

Response: Please see opening remarks. Again, we agree with this assessment. Our new data suggests that both Rac1 and RhoA activity are elevated following Rab35 knockdown (**Fig. 8; S11**). We assert that this chronic activation of actin remodeling is the mechanism underlying dysmorphic sprout formation, loss of cell polarity as well as shifts in the cellular globular and filamentous actin content.

Other specific points:

1. Although the authors stated in the text that they analyzed known Rab35 effectors by a holistic approach, they tested only three Rab35 effectors, ACAP2, OCRL, and MICAL-L1. Two other important actin-regulating Rab35 effectors, MICAL1 (Nat. Commun. 2017;8:14528) and Fascin (Science 2009;325:1250-1254), are missing in this manuscript. The authors should test their involvement in angiogenesis.

Response: In the revised manuscript we have tested both MICAL1 and Fascin effectors as well. Neither effector fully recapitulated the Rab35 phenotype (*please see new data Fig. 3, S6, S7*).

2. Important control experiments are missing in Fig. 2C. The authors should also test Scram+Rab35-CA and Scram+Rab35-DN.

Response: We have performed these controls (*see new data Fig. 2D*).

3. Knockdown of Rab35 effectors shown in Fig. 3B is not so efficient, in comparison with the Rab35 knockdown (Fig. 1C). This raises the possibility that insufficient knockdown of Rab35 effectors in HUVECs would mask phenotypes in angiogenesis. The authors need to evaluate the knockdown phenotypes by using more effective siRNAs. Also, knockdown efficiencies in Fig. 1C, Fig. 3B, Fig. 4B, and Fig. S4F should be calculated.

Response: We have repeated these experiments and blots using a more effective siRNA and have included the knockdown efficiencies in the figure legends based on three independent repeats (**Fig. 3B**).

4. Based on the results shown in Fig. S7D and E, the authors suggested that ACAP2 does not interact with or act on Arf6. However, ACAP2 is an enzyme (Arf6-GTPase activating protein) and does not need to stably interact or colocalize with its substrate. The authors should consider this point.

Response: We have considered this point and new data for a Rab35/ACAP2/Arf6 interaction is presented (**Fig. S8, S9**), we also now mention this point (*see lines 299-300*).

5. Arf6 overexpression phenotypes (Fig. S7H) seem to be opposite to Rab35 overexpression phenotypes (Fig. 2C). Rab35-DN increased non-lumenized sprouts, whereas Arf6-DN decreased them, suggesting that both Rab35 and Arf6 are involved in blood vessel development. Although the authors claimed that the level of active Arf6 was not changed even after Rab35 depletion or overexpression without showing quantitative data, this reviewer think that inactivation of Arf6 would occur locally (e.g., at the apical membrane alone). To address this issue, effect of plasma membrane-targeted ACAP2 on angiogenesis should be tested.

Response: We did not construct a new membrane-targeted ACAP2 as this protein is already primarily localized to the plasma membrane (**Fig. 4A**). However, we have included a new figure demonstrating how Arf6 is indeed important for sprout morphodynamics (*see new data, Fig. S9*) as others have reported [4]. New data in the manuscript now demonstrates that

loss of Rab35 does significantly activate Arf6; however, Rab35's effect on sprouting angiogenesis is likely not reliant on this pathway.

6. In Fig. 4B, knockdown efficiency of DENNd1a-c should be investigated by Western blotting, the same as in Fig. S8B. Mr size of DENNd1a-c should also be provided in Fig. S8B.

Response: The reason for using RT-PCR for validating KD efficiency is that we ran a series of controls with commercial antibodies and could not discriminate between the DENNd1 family members (1A-C), or other distantly related DENN proteins (e.g. DENNd4a); although, these antibodies were touted to be specific. Because of this, we are more confident in our use of specific mRNA primers to determine knockdown efficiencies between the DENN1A-C family members.

7. Other minor points:

- The manuscript contains a lot of typos. The authors should carefully check the manuscript. e.g., drosophila  Drosophila (line 71), 37°C degree  37°C (line 218), Laria-Bertani  Luria-Bertani (line 246), E Coli  E. coli (line 247), MICAL-L1  MICAL1 (line 561), and RUSC  RUSC2 (lines 412, 641 and 644).

- Information about DENNd1a-c antibodies is missing in Resource Table.

Response: all errors have been addressed.

Reviewer #2 (Remarks to the Author):

This interesting manuscript by Francis et. al explores the unexplored roles of Rab35 in sprouting angiogenesis. The authors apply a combination of in vitro and in vivo tools to suggest how this small GTPase modulates cytoskeletal events underpinning vessel formation. In this context, the authors argue that Rab35 plays its most critical role in apicobasal polarity with severe implications in lumenogenesis. They show that the GEF DENNd1c does not activate Rab35 in endothelial cells, but recruits it to sites of actin polymerization. This is a very elegant study with robust data and sound experimental controls. Overall, the manuscript is very well presented, clear and well written. Nevertheless, an overall sense of lack of novelty runs across the manuscript. Although the authors claim “a novel scenario that trafficking-based regulators can control vital crosstalk with the actin cytoskeleton”, this has been shown elsewhere. For example, the McPherson and Scott labs have linked RAB35 to Fascin, whilst Sasaki and colleagues have done extensive work to dissect the links between RAB13 and Filamin activity. Furthermore, several points could be addressed to make the current findings more compelling.

Response: Thank you for the kind words. With regard to novelty, please see our opening statement. More specifically, the McPherson and Scott labs body of literature agrees with our finding that Rab35 directly affects actin; however, beyond demonstrating an association with actin dynamics or targeting through its GEF, there is little other mechanistic information. In the revised manuscript we have attempted to reproduce Rab35's association with Fascin in ECs, but did not find that Fascin recapitulated the Rab35 phenotype (*please see new data Fig. 3, S6, S7*). Again, our new data demonstrates a regulatory role of Rab35 in modulating both Rac1 and RhoA activity, which explains Rab35's drastic effect on actin architecture, polarity signaling and downstream sprouting morphology.

Major comments:

1. The role of Rab35 in cell migration is widely overlooked. Rab35 loss-of-function in the fibrin assays shows sprouts that “appeared stubby”, are shorter and less abundant than in control cultures. (Figure 1D-F). Given the cell motility roles attributed to Rab35 in other systems, it is likely that Rab35 modulates endothelial migration steps preceding lumen formation, explaining the aforementioned loss-of-function phenotypes. The authors then claim that no cell motility

defects are observed in Fig. S5A, but they do not carry out a comprehensive analysis of motility parameters. Of note, these experiments are performed in 2D and some caution must be taken when extrapolating to 3D scenarios.

Response: We contend that this is a challenging question to isolate in 3D sprouting scenarios as migration and polarity signaling are inexorably intertwined. For instance, loss of junctional, apical trafficking, or metabolic proteins can affect sprouting parameters. The point being, that cell migration in 3D sprouting is functionally linked to other, non-motility-based, signaling. Because of this, we believed our best course of action was to first answer how Rab35 could influence migration, per se. To do so, we monitored collective cell migration (scratch wound) in 2D (**Fig. S6**) and did not find any significant differences between groups. Secondly, and more indirect, we quantified how Rab35 influenced cell protrusive and retractive membrane dynamics, which have a direct bearing on 2D cell motility behaviors. Here, we were also surprised to how little the Rab35 knockdown affected membrane dynamics in reference to its devastating effects on 3D sprouting. To better answer, your specific question, we created mosaic sprouts (50:50, Scramble:Rab35 KD) and tracked the position of individual ECs during sprout growth. The prediction would be if Rab35 significantly reduced cell motility then a greater number of the Rab35 knockdown ECs would remain on the bead, or in the stalk, opposed to the tip-cell position. Our data indicated no difference between control and loss of Rab35 in terms of EC positioning (see new data **Fig. S4**). Given we had the same basic outcome using three independent tests for cell motility, we concluded that the major issue underpinning sprouting-related abnormalities were downstream of altered polarity signaling. This is also congruent with the mis-localization of polarity markers in the absence of Rab35 (**Fig. S3**).

2. If indeed Rab35 only plays critical roles in endothelial cell apicobasal polarity (to control lumen formation and not exerting effects in endothelial migration), did the authors explore if loss of Rab35 affects polarity in quiescent endothelial cells prior to sprouting? Could the perturbed apicobasal polarity impair cytoskeleton arrangements during tip cell selection?

Response: Our current data indicates that loss of Rab35 does not impede tip-cell positioning (**Fig. S4C**), or motility parameters in general. Additionally, individual ECs that lack Rab35 demonstrate localized lumen collapse (**Fig. 1I,J,M**). Given this data, we believe it is doubtful that loss of Rab35 impedes tip-cell selection, although our results would suggest alterations in the cytoskeleton and polarity. This Rab35-mediated cytoskeletal dysfunction most significantly manifests when apicobasal polarity is required, such as in lumen formation.

We have not explored how loss of Rab35 affects quiescent ECs as our goal was to relate perturbations in Rab35 to angiogenic processes. However, this is a great point, but a tricky parameter to accurately test. We previously reported that ECs cultured in 2D do not show apicobasal polarity (Francis et al. *Microcirculation*. 2021). In this context it would not be possible to knockdown Rab35 in a quiescent monolayer and then determine podocalyxin location (for example) as this polarity axis is non-existent outside a 3D sprout structure.

3. In line with the prior comments, it is relevant to be clear about the role of Rab35 in tip cells. The authors disregard the topic mentioning that Rab35 “had no preference for tip cell” positioning. It is not very clear how they infer this without 3D imaging.

Response: Yes, our results strongly support a cell autonomous role for Rab35 in which knockdown of Rab35 neither increases or decreases the propensity of an EC to become a tip-cell. Again, we have confirmed this using mosaic experiments in sprouts (**Fig. S4A-C**). This result is also in-line with the notion of cell motility being largely intact despite the loss of Rab35. As for imaging, our lab is very adept at sub-cellular imaging in the fibrin-bead sprout model. We routinely image individual cells, membrane domains and organelle compartments in the sprout collective (Francis et al. *ATVB*. 2021; Francis et al. *Microcirc*. 2021). As such, we are confident that our quantification of cell position (tip vs. stalk) in a sprout is accurate. To this point, in

quantifying position, we always collect z-stacks to ensure the entire cell is being scrutinized-these could be used to make a 3D rendering. However, we typically present a single cross-sectional slice to best highlight the luminal cavity or membrane domains.

4. The authors demonstrate that RAB35 does not act via the ACAP2-Arf6 axis, but do not look for other potential effectors in endothelial cells that could explain the modulation of actin. Could Rab35 regulate Fascin? The material and methods section indicates the use of a Fascin inhibitor and Fascin plasmid but to my knowledge, no data in the manuscript reflects this.

Response: We have re-examined the interplay between Arf6 and Rab35. Our initial data did not suggest an interplay. After using several commercial kits for determining Arf6 activity, we did detect a significant increase in Arf6 activity with Rab35 knockdown as previously reported (**Fig. S9B**). To this end, our conclusions have not changed in showing that aberrant Arf6 activation is largely not harmful to sprout formation; thus, still an unlikely cause for dysmorphic sprouts in the absence of Rab35 (**Fig. S9D**). In the revised manuscript, we have explored Fascin (please see new data, **Fig. 3,4, S6, S7**).

5. The in vivo work is interesting and very relevant. Although the vital role played by Rab35 may explain the major developmental defects observed in CRISPR mutants, it is critical to demonstrate that the CRISPR guides do not have potential off-targets.

Response: This is an important point. Our methods are identical to those published by Hoshijima et al. (Dev Cell. 2019) in which potential off target effects were thoroughly screened for and not detected. Using this method, we employ commercially synthesized guides, Cas9 engineered for specificity, and a guide algorithm to reduced off-target editing as detailed by the aforementioned paper. In the revised manuscript, we provide the top potential off-target loci all of which are intronic, except for one guide only when factoring-in for two nucleotide mismatches (**Fig. S11**). Given single mismatches are all intronic, even if wrongfully edited, off-target indels would not likely cause a phenotype as no coding regions would be mutated. We hope that this is sufficient as this is an established method and performing whole genome sequencing to validate our approach seems like a bit of an overkill given the depth of literature on this technology.

6. The use of a Rab35 biosensor to prove exactly where this small GTPase is active during spouting angiogenesis would significantly substantiate the findings.

Response: Our data strongly advocates that Rab35 is cell autonomous and is largely affiliated with actin regulation. As such, we contend it would be unlikely there would be a discernable sprout-wide activity pattern as Rab35 would be participating in the regulation of actin assembly on the individual cell basis. Our data using mutants clearly depicts how the active and inactive forms of Rab35 localize in sprouts and different sprout locations (**Fig. S3A, S4A-C**), subcellularly in sprouts (**Fig. 1A; 2B,D; 5C; 8M; S2A,B; S9J**), statically in 2D (**Fig. 4A; 6; S3C-D,S8A; S10B,G**), or dynamically in 2D (**Movies 6-9; Fig. 6**). We do agree this experiment would provide valuable data on the biological regulation of Rab35, but we feel is slightly outside the scope of our characterization goals. In the future, we would very much like to construct and validate such a tool to better understand how Rab35 works regardless of its role in sprouting.

Minor comments:

1. Because siRNAs drop efficiency over time and thus, it would be useful to indicate the stages of sprout analysis and Western blotting. It is not clear from the methods section how long the fibrin sprouts are cultured or when the siRNA efficiencies are confirmed.

Response: We agree this is an important control. To address this, we monitored knockdown efficiency in 2D culture, which has a substantially higher proliferation rate as compared with cells in fibrin-bead sprouts. We found that at day 4, the day in which sprout are

assayed, the knockdown efficiency was >70% (**Fig. S1D**). Given sprouts proliferate significantly less than cells cultured on a dish, it can be assumed less siRNA would be lost overtime and knockdown efficiency would be acceptably high.

2. Why did the authors present error bars in the form of 95% confidence intervals? Although valid, this is harder to interpret than standard deviation.

Response: We have changed all graphs to standard deviation.

3. In line 367 the authors mention that Rab35 is necessary for sprout function. I don't think that this is proven at this stage (although this is later on shown with in vivo data).

Response: this line has been changed (See line 215).

4. In Figure 3B, the siRNA knockdown of Rab35 effectors is not very efficient. Could loss of a small amount of protein justify the phenotypes? Could the authors try other siRNAs?

Response: we have redone noted experiments, immunoblots and calculated knockdown efficiencies which are now included in the revised figure legends.

5. The authors state "Arp2 was no longer located on actin" in line 530, but there is no actin marker/staining in Figure 5E.

Response: this line has been changed (see 363).

6. The authors mention that Rab35-CA alters protrusions formation and retraction "in line with enhanced migration". Where is enhanced migration shown?

Response: This refers to previous literature regarding this topic. This line has been changed and referenced (see line 392).

7. Some plasmids are missing in material and methods.

Response: Due to space limitations the methods are truncated. However, all reagents are now listed in the supplemental information.

LITERATURE CITED

1. Zhu, Y., et al., *Rab35 is required for Wnt5a/Dvl2-induced Rac1 activation and cell migration in MCF-7 breast cancer cells*. Cell Signal, 2013. **25**(5): p. 1075-85.
2. Shim, J., et al., *Rab35 mediates transport of Cdc42 and Rac1 to the plasma membrane during phagocytosis*. Mol Cell Biol, 2010. **30**(6): p. 1421-33.
3. Chevallier, J., et al., *Rab35 regulates neurite outgrowth and cell shape*. FEBS Lett, 2009. **583**(7): p. 1096-101.
4. Davis, C.T., et al., *ARF6 inhibition stabilizes the vasculature and enhances survival during endotoxic shock*. J Immunol, 2014. **192**(12): p. 6045-52.

REVIEWER COMMENTS

Reviewer #1 (Remarks to the Author):

In the revised manuscript, the authors showed additional evidence that the six known Rab35 effectors are not directly involved in Rab35-mediated sprout formation by endothelial cells. They also unexpectedly found that Rab35 depletion increased both Rac1 and RhoA activities, which consequently promote actin rearrangement in dysmorphic sprouts. While the underlying mechanism by which Rab35 limits Rac1 and RhoA activities is still completely unknown, these new findings would greatly accelerate our understanding of the role of Rab35-mediated membrane trafficking in blood vessel development.

One minor point:

(Fig. S9E) "KD of Rab35" should be "KD of Arf6".

Reviewer #2 (Remarks to the Author):

The authors have addressed all points and where necessary, provided new data or made changes that improved the manuscript. Nevertheless, a couple of points must be addressed:

1. In the new version of the manuscript, the authors claim that new siRNAs targeting Rab35 effectors show greater KD efficiencies than those presented in the original manuscript. This seems to be the case as WB data is used as evidence (Figure 3B). However, some of the datapoints and "n" in phenotypic analyses of the sprouts (Figure 3E and F) have not changed from the original version of the manuscript to the updated one. For example, the vessel length measurements and non-lum. vessel counting in ACAP2 and OCRL siRNA experiments are exactly the same in both versions. Would you not expect some level of technical and biological variability in your quantifications in independent experiments with new siRNAs?

2. In their answer to my major comment 6 (first round of revisions) regarding potential CRISPR off targets, the authors state: "single mismatches are all intronic, even if wrongfully edited, off-target indels would not likely cause a phenotype as no coding regions would be mutated". This is a rather bold statement, as indels in regulatory regions contained within introns may result in phenotypes that could be as severe as in coding regions. Although I agree that whole genome sequencing to validate their approach may be an "overkill", given the small number of predicted off targets, targeted sequencing of these regions should not be a cumbersome task.

Reviewer #1 (Remarks to the Author):

In the revised manuscript, the authors showed additional evidence that the six known Rab35 effectors are not directly involved in Rab35-mediated sprout formation by endothelial cells. They also unexpectedly found that Rab35 depletion increased both Rac1 and RhoA activities, which consequently promote actin rearrangement in dysmorphic sprouts. While the underlying mechanism by which Rab35 limits Rac1 and RhoA activities is still completely unknown, these new findings would greatly accelerate our understanding of the role of Rab35-mediated membrane trafficking in blood vessel development.

One minor point:

(Fig. S9E) “KD of Rab35” should be “KD of Arf6”.

Response: Thank you for finding this error, it has been fixed.

Reviewer #2 (Remarks to the Author):

The authors have addressed all points and where necessary, provided new data or made changes that improved the manuscript. Nevertheless, a couple of points must be addressed:

1. In the new version of the manuscript, the authors claim that new siRNAs targeting Rab35 effectors show greater KD efficiencies than those presented in the original manuscript. This seems to be the case as WB data is used as evidence (Figure 3B). However, some of the datapoints and "n" in phenotypic analyses of the sprouts (Figure 3E and F) have not changed from the original version of the manuscript to the updated one. For example, the vessel length measurements and non-lum. vessel counting in ACAP2 and OCRL siRNA experiments are exactly the same in both versions. Wo

uld you not expect some level of technical and biological variability in your quantifications in independent experiments with new siRNAs?

Response: Thank you for finding this error, we neglected to update the graphs with the new data. In the revised manuscript the data for Fig. 3E,F has been updated and no conclusions were changed.

2. In their answer to my major comment 6 (first round of revisions) regarding potential CRISPR off targets, the authors state: “single mismatches are all intronic, even if wrongfully edited, off-target indels would not likely cause a phenotype as no coding regions would be mutated”. This is a rather bold statement, as indels in regulatory regions contained within introns may result in phenotypes that could be as severe as in coding regions. Although I agree that whole genome sequencing to validate their approach may be an “overkill”, given the small number of predicted off targets, targeted sequencing of these regions should not be a cumbersome task.

Response: We agree that mutations of intronic regions can also produce significant phenotypes. We sequenced the only off-target site in close proximity to a coding region and did not detect any sequence differences (see associated figure).

Score	Expect	Identities	Gaps	Strand
2547 bits(1379)	0.0	1379/1379(100%)	0/1379(0%)	Plus/Plus
Query 3	TTTGATTACACAAGACAGGTAACAGAAAAACACAATTCATTTTGTTCGCAATTATTTCCC	62		
Sbjct 1	TTTGATTACACAAGACAGGTAACAGAAAAACACAATTCATTTTGTTCGCAATTATTTCCC	60		
Query 63	ACCCCAACAAAAGAACATCACTCTCTTCTCCCTATACAAATACACAGAATTCCTTACA	122		
Sbjct 61	ACCCCAACAAAAGAACATCACTCTCTTCTCCCTATACAAATACACAGAATTCCTTACA	120		
Query 123	AGAGACAATGTAATGCAAAAACAAATATAGTTTGGTACCCAAAAACCAAGTATACATTAC	182		
Sbjct 121	AGAGACAATGTAATGCAAAAACAAATATAGTTTGGTACCCAAAAACCAAGTATACATTAC	180		
Query 183	TGTACTTTACTAATAAAAAGTTACATTATCcaacaaaaaaaatgaaacacacacacaaaa	242		
Sbjct 181	TGTACTTTACTAATAAAAAGTTACATTATCcaacaaaaaaaatgaaacacacacacaaaa	240		
Query 243	ataaataaaataaaaaaacacTATACTTAGTTACACTTCTGTAAATAATGACCAATAT	302		
Sbjct 241	ATAAATAAATAAAAAAACACTATACTTAGTTACACTTCTGTAAATAATGACCAATAT	300		
Query 303	TATTGTTATTAAATGAGTAATTAACGGCTTCCAAATTTCTTCATATTTATCCATTTGGT	362		
Sbjct 301	TATTGTTATTAAATGAGTAATTAACGGCTTCCAAATTTCTTCATATTTATCCATTTGGT	360		
Query 363	TTTTAATAAAATATGTTACTTTTTCAATGGCCATACACAGTGGCATTGATTAATCCAAG	422		
Sbjct 361	TTTTAATAAAATATGTTACTTTTTCAATGGCCATACACAGTGGCATTGATTAATCCAAG	420		
Query 423	TTCCAGTAGATGGTGGATCAGGCTTTTTCCAGCGATCATTCTTTTAGCTTGTAGTATAGC	482		
Sbjct 421	TTCCAGTAGATGGTGGATCAGGCTTTTTCCAGCGATCATTCTTTTAGCTTGTAGTATAGC	480		
Query 483	ATAATCTATTAATTTGCGCTGCTGGAGATTTAGTTAAGGTTCTTTGGGTATAAATGTAG	542		
Sbjct 481	ATAATCTATTAATTTGCGCTGCTGGAGATTTAGTTAAGGTTCTTTGGGTATAAATGTAG	540		
Query 543	CAAAATCATTTTTGCATCTAATGGAATATTCAAGGGTACAATATCTTTGTAATTAAT	602		
Sbjct 541	CAAAATCATTTTTGCATCTAATGGAATATTCAAGGGTACAATATCTTTGTAATTAAT	600		
Query 603	CACCTTACCCAAAAGTCTCTAATAATGTTGCATTCACAAACACAATGCACACATGAACC	662		
Sbjct 601	CACCTTACCCAAAAGTCTCTAATAATGTTGCATTCACAAACACAATGCACACATGAACC	660		
Query 663	CTGAGCCTCTCTGCATTTAAAGCATGTATCAGGAATATTTTTATTAACCTATTTAGTTT	722		
Sbjct 661	CTGAGCCTCTCTGCATTTAAAGCATGTATCAGGAATATTTTTATTAACCTATTTAGTTT	720		
Query 723	GACAGGAGTTATATAAGTCTGCATCAACCAGTTATATTGAAGTAGTTTAAAGCTAACGCT	782		
Sbjct 721	GACAGGAGTTATATAAGTCTGCATCAACCAGTTATATTGAAGTAGTTTAAAGCTAACGCT	780		
Query 783	CCCTAACTGTTCTGAGAATTTTACAGAATTGTCTTCATTATCCTCAGATATATCCTC	842		
Sbjct 781	CCCTAACTGTTCTGAGAATTTTACAGAATTGTCTTCATTATCCTCAGATATATCCTC	840		
Query 843	ATTCAATCCCATCTTCCAAATTTCAAGTTTTGGTGTGATGAATCAGTAGACCCGTATAC	902		
Sbjct 841	ATTCAATCCCATCTTCCAAATTTCAAGTTTTGGTGTGATGAATCAGTAGACCCGTATAC	900		
Query 903	AAGCCAGTTGTAAAAAACTGATATTAACCCCTTATTATTGCAACTTTGGTAATAAATTT	962		
Sbjct 901	AAGCCAGTTGTAAAAAACTGATATTAACCCCTTATTATTGCAACTTTGGTAATAAATTT	960		
Query 963	TTCCAAAATTTAAAGTACTGGCAAGTTTATTGATTGTTTCTGCCTAGAAAGAACAAAAC	1022		
Sbjct 961	TTCCAAAATTTAAAGTACTGGCAAGTTTATTGATTGTTTCTGCCTAGAAAGAACAAAAC	1020		
Query 1023	CCTTATCTGTAAGTATTTAAAGAAATGCTTTTTTGGAAATTTGACTTAATAGATAATTG	1082		
Sbjct 1021	CCTTATCTGTAAGTATTTAAAGAAATGCTTTTTTGGAAATTTGACTTAATAGATAATTG	1080		
Query 1083	CTCAAACAACATCAACGTTCCCTCCTAATAGATCACTAATTTTGCTTAATCCCTTTTC	1142		
Sbjct 1081	CTCAAACAACATCAACGTTCCCTCCTAATAGATCACTAATTTTGCTTAATCCCTTTTC	1140		

Figure 1. Alignment of scramble (Query) and Rab35-A (Subject) CRISPR guide injected zebrafish DNA sequences flanking predicted exonic target site with 2 mismatches.

REVIEWERS' COMMENTS

Reviewer #2 (Remarks to the Author):

All points have now been addressed and this final version of the manuscript is a significant improvement from the original one.